# CAPO: Towards Enhancing LLM Reasoning through Generative Credit Assignment

## Abstract

Reinforcement Learning with Verifiable Rewards (RLVR) has improved the reasoning abilities of Large Language Models (LLMs) by using rule-based binary feedback, helping to mitigate reward hacking. However, current RLVR methods typically treat whole responses as single actions, assigning the same reward to every token. This coarse-grained feedback hampers precise credit assignment, making it hard for models to identify which reasoning steps lead to success or failure, and often results in suboptimal policies. Methods like PPO provide credit assignment by value estimation, but yield inaccurate and unverifiable signals due to limited sampling. On the other hand, methods using Process Reward Models can provide step-wise rewards but suffer from several key limitations: they require high-quality process supervision labels, the feedback is unreliable due to probabilistic reward modeling, and their application in online reinforcement learning (RL) is time-consuming. To overcome these limitations, we introduce a simple but efficient method—Credit Assignment Policy Optimization (CAPO). CAPO avoids the complexities of prior approaches. Instead of training auxiliary models, CAPO directly leverages an off-the-shelf, general-purpose LLM as a Generative Process Reward Model (LLM-as-GenPRM) to generate all step-wise critique by one pass only based on the correctness of the step itself, providing deterministic token-level credits to refine the tokens that were originally assigned identical rule-based rewards. This design choice not only simplifies the training pipeline but also enhances its generality, as our experiments show it works effectively with various powerful, widely accessible open-source models. The fine-grained feedback enables a crucial shift from purely outcome-oriented to process-oriented learning; our analysis of this dynamic leads to a reward structure that balances both objectives. To further enhance the accuracy and robustness, we employ voting mechanisms that scale with the number of generated critiques. Extensive experiments on various backbones like Llama and Qwen models show that CAPO consistently outperforms supervised learning-based and RL-based fine-tuning methods across four challenging mathematical benchmarks and three out-of-domain benchmarks. Further analysis shows that CAPO can help the model to foster the learning of correct reasoning pathways leading to correct answers.

## 1 Introduction

Reinforcement Learning with Verifiable Rewards (RLVR) has demonstrated significant success in enhancing the reasoning capabilities of Large Language Models (LLMs) (Lambert et al., 2024; Guo et al., 2025). Particularly in the post-training phase, RLVR enhances models' mathematical and code reasoning abilities. Recently, a substantial body of work (Su et al., 2025; Lu, 2025; Sane, 2025; Dao & Vu, 2025; Tang et al., 2025b) has extended the application of RLVR to diverse domains beyond mathematics and code, achieving widespread validation and notable progress. RLVR employs verifiable, rule-based reward functions to provide models with clear binary feedback during reinforcement learning. This approach mitigates reward-hacking, often caused by reliance on subjective human evaluations or complex reward models (Eisenstein et al., 2023; Gao et al., 2023; Dong et al., 2023), thereby fostering a more transparent and efficient training process.

However, prevailing rule-based RLVR methods typically assign the binary reward of the entire response as the raw return to all tokens. *This approach leads to two significant issues: it fails to provide differentiated feedback and entirely neglects the problem of credit assignment.* Firstly, we

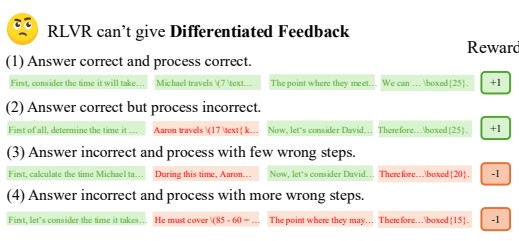
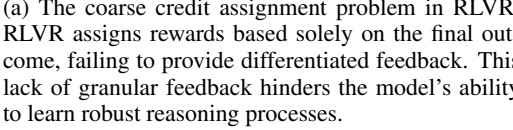

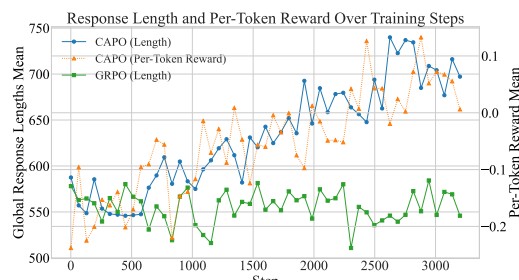

(a) The coarse credit assignment problem in RLVR. RLVR assigns rewards based solely on the final outcome, failing to provide differentiated feedback. This lack of granular feedback hinders the model's ability to learn robust reasoning processes.

(b) Training dynamics of CAPO on Qwen2.5-7B. The growth in CAPO's response length indicates *effective exploration*, where the model learns to produce more meaningful, exploratory steps, leading to higher accuracy.

Figure 1: Addressing the coarse credit assignment problem with CAPO. (a) A schematic illustrating the core limitation of RLVR, where a single final reward fails to provide granular feedback for the reasoning process. (b) The training dynamics of our CAPO, on Qwen2.5-7B, demonstrating effective learning where increased exploration (longer responses) correlates with higher final accuracy.

use an example to illustrate the problem of lack of differentiated feedback in Figure 1a. The quality of different responses to a single question can vary dramatically. A significant gap can exist even among responses that are all correct or all incorrect. As shown in Figure 1a, a response with a correct answer might have a completely correct process or a few containing flawed steps. Similarly, a response with an incorrect answer might have a process that is mostly wrong, or one where only a few steps are erroneous. Therefore, merely assigning a binary reward is far from sufficient. Secondly, the current rule-based RLVR methods simplified the RL problem to a contextual dueling bandit setting (Dudík et al., 2015; Zeng et al., 2024; Yue et al., 2012) where the entire response is treated as a single action, thereby avoiding the need for a value model. And therefore it ignored the *credit assignment* step, which is an important step in reinforcement learning.

On the other hand, existing methods providing fine-grained credit are prone to reward hacking. As shown in Table 1, in addition to PPO, to conduct fine-grained credit assignment, some works densify the rewards by decomposing sequence-level rewards to token-level feedback, using attention-based credit (Chan et al., 2024) or Shapley values (Cao et al., 2025) (densification-based RL). However, these methods rely on a reward model or other complicated external sources, causing potential reward hacking. Another line of work employs process reward models (PRMs) (Lightman et al., 2023; Setlur et al., 2024). While PRMs can provide step-wise rewards, they face significant practical challenges: they rely on high-quality, costly manual annotations and require multiple inference calls for each step, reducing their efficiency in online RL. Beyond these practical issues, the core problem with PRMs and estimation-based methods mentioned above still lies in the inaccuracy and unreliability of their credit assignment. The value model in PPO, for instance, is unreliable due to its dual-optimization objective. Similarly, existing PRMs typically model process rewards as a form of value estimation, predicting the future returns of a current state. However, this Monte Carlo-based estimation is challenging due to limited sampling and the intricate interaction between the policy and completion models. Worse still, optimizing PRMs on the Best-of-N evaluation metric often causes an unintended "process-to-outcome" shift (Zhang et al., 2025c), where the model learns to prioritize the final answer over a sound reasoning process, undermining the purpose of process supervision.

To address these challenges, we introduce Credit Assignment Policy Optimization (CAPO). Inspired by GRPO's simplicity, we discard the probabilistic value estimation of process rewards and adopt a deterministic, binary evaluation. Our approach focuses solely on the intrinsic correctness of each step—identifying objective errors like flawed calculations or logic—rather than predicting its long-term value. This simpler, more straightforward approach reduces the risk of reward hacking. Specifically, we use an off-the-shelf, general-purpose LLM as a Generative Process Reward Model (LLM-as-GenPRM) to achieve this goal, prompting it to efficiently generate all step-wise critique(or said judgment) in a single pass only based on the correctness of the step itself, thereby providing step-wise credit feedback for each generated response during online RL. This makes our method

Table 1: Comparison of existing methods to conduct credit assignment. Here $N$ is a number larger than 1. CAPO is an efficient, simple, and general method that provides reliable credit assignment.

| Method | High-Quality Data | Number of Inference Calls | Providing Reliable Rewards | Test-Time Scaling | Fine-grained Credits |
|---|---|---|---|---|---|
| PPO | × | 1 | × | × | ✓ |
| Densification-based RL | × | 1 | × | × | ✓ |
| PRM | ✓ | N | × | × | ✓ |
| CAPO | × | 1 | ✓ | ✓ | ✓ |

remarkably simple to implement and broadly applicable, as it does not require specialized or finely-tuned reward models. The use of publicly available, powerful LLMs as the GenPRM significantly lowers the barrier for replication and further extension by the research community. We then apply a penalty to the credits of their corresponding tokens that were originally assigned identical rule-based rewards. This enables a shift from outcome- to process-oriented learning, but we find this introduces a new challenge: a conflict between the two reward signals. We therefore analyze their dynamics and utilize reward shaping to create a hierarchical reward structure, effectively balancing both objectives. To enhance robustness and accuracy, we scale up the number of critiques generated by the LLM-as-GenPRM and use voting mechanisms to produce more accurate penalty assignments. Finally, we use these credits to conduct policy optimization.

The extensive experiments show that CAPO consistently outperforms both supervised fine-tuning and reinforcement learning-based methods. Furthermore, our analysis reveals the mechanism behind this success: CAPO actively fosters the learning of correct reasoning pathways, guiding the model to generate solution trajectories that lead to correct answers and promote more effective exploration.

We summary our main contributions as follow:

- We identify the critical challenge in RL-based LLM fine-tuning, where current methods fail to achieve fine granularity, reliability, and efficiency simultaneously. We show that while standard RLVR is reliable, it lacks granularity. Conversely, fine-grained methods often become unreliable and susceptible to reward hacking due to their reliance on complex, estimation-based signals.

- We propose Credit Assignment Policy Optimization (CAPO), an efficient and simple method for the critical credit assignment problem in RLVR. We propose to focus back on the correctness of the step itself and employ an LLM-as-GenPRM to assign reliable step-wise rewards, enabling token-level fine-grained credit attribution.

- We conduct an in-depth analysis of the shift from outcome- to process-oriented learning enabled by fine-grained rewards. We identify a critical conflict between these two signals and, based on our empirical observations, propose an asymmetric reward shaping strategy. Our approach creates a hierarchical learning objective that prioritizes correct outcomes while secondarily encouraging sound reasoning, effectively balancing the two signals.

- Extensive experiments on four mathematical and three general reasoning benchmarks with Llama3-1B/3B and Qwen2.5-1.5B/7B/14B show that CAPO consistently outperforms supervised learning and other RL approaches lacking fine-grained credit assignment.

## 2 PROBLEM ANALYSES

Reinforcement Learning provides a robust paradigm for fine-tuning Large Language Models (LLMs) beyond standard supervised objectives (Ziegler et al., 2019; Ouyang et al., 2022). In this paradigm, the LLM acts as a policy $\pi_\theta$ that sequentially generates tokens $a_t$ (actions) given the preceding sequence $s_t$ (state), forming a trajectory $\tau = (s_0, a_0, \ldots, s_T, a_T)$ where the state $s_t$ typically represents the sequence of previously generated tokens (e.g., $o_0, \ldots, o_{t-1}$), and an action $a_t$ is the selection of the next token $o_t$ from the vocabulary $\mathcal{V}$. The goal is to optimize $\pi_\theta$ to maximize the expected cumulative reward $J(\theta) = \mathbb{E}_{\tau \sim \pi_\theta}[\sum_{t=0}^T \gamma^t r_t]$.

While the framework is well-defined, a critical and unresolved challenge lies in credit assignment: determining which specific actions within a long trajectory are responsible for the final outcome and

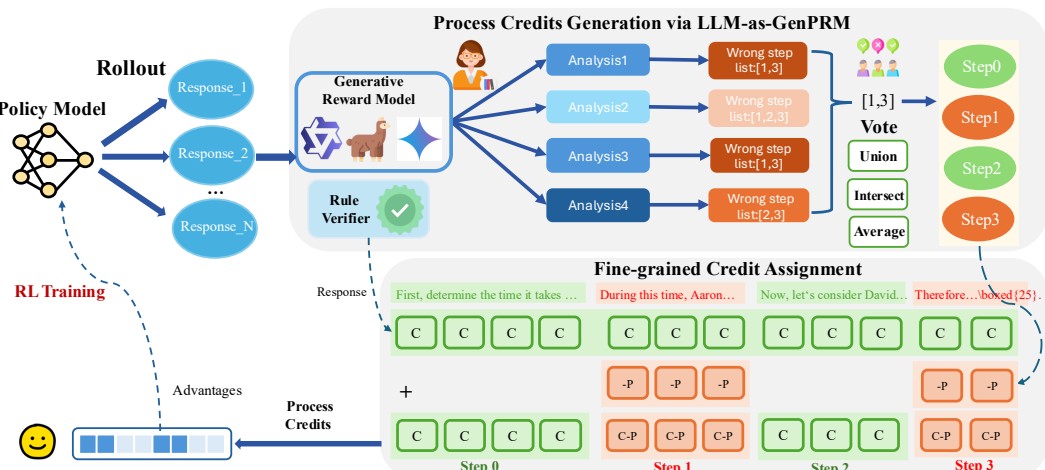

Figure 2: An overview of our method Credit Assignment Policy Optimization (CAPO). We utilize a LLM as a GenPRM to identify all incorrect steps within a model's generated rollout in a single pass. During credit assignment, we then suppress these erroneous steps, which prevents the correct portions of the sequence from being unfairly penalized, enabling the model to learn correct reasoning pathways. We denote $W_{\text{whole}}$ and $W_{\text{process}}$ as C and P for short.

to what extent. Below, we analyze the limitations of prevailing credit assignment strategies, which motivates our work.

**The Coarse-Grained Feedback Dilemma in Outcome-Based RL.** A typical strategy in RLVR, exemplified by methods like Group Relative Policy Optimization (GRPO) (Shao et al., 2024), is to assign an identical reward to all tokens based solely on the final outcome as $\hat{A}_t^i = \frac{R^i - \text{mean}(\{R^i\}_{i=1}^G)}{\text{std}(\{R^i\}_{i=1}^G)}$ for any token $o_t$, $t = \{1, \ldots, T\}$, where $R^i$ is the response-level rule-based reward. The absence of granular feedback in GRPO means the model cannot differentiate between partially correct and entirely flawed reasoning. This makes it difficult to learn robust reasoning logic, as the model fails to reinforce correct steps or penalize individual errors.

**The Vulnerability to Reward Hacking in Fine-Grained Approaches.** We argue that the primary challenge in achieving fine-grained credit assignment lies in providing reliable process credits while avoiding reward hacking. A key factor in the success of methods like GRPO is their simple and direct reward design, which minimizes dependence on external sources (such as auxiliary models or complex rules) and is thus robust against reward hacking. In contrast, while methods utilizing an auxiliary model—like a reward model (Chan et al., 2024) or a value model (Schulman et al., 2017)—can offer fine-grained credit assignment, they are inherently vulnerable to being hacked by this very model. The policy can learn to exploit inaccuracies or biases in the auxiliary model to maximize rewards, often at the expense of true performance. This vulnerability also extends to approaches based on PRMs. Beyond suffering from the high costs of online RL application and manual annotations, PRMs face a more fundamental issue. They typically function as predictive estimators of a solution's potential, often relying on Monte Carlo methods Such probabilistic modeling can introduce substantial noise and unreliability into the process-level rewards (Zhang et al., 2025c), creating opportunities for the policy to find and exploit spurious correlations.

Therefore, a critical research question emerges: how can we design a credit assignment method that is both simple and effective, providing the benefits of fine-grained feedback without introducing complex dependencies that lead to reward hacking?

## 3 CAPO: THE PROPOSED APPROACH

The preceding analysis of existing methodologies reveals a critical gap in the current works of RL-based LLM fine-tuning. We therefore propose our approach, called Credit Assignment Policy Optimization (CAPO), which aims to achieve fine-grained and reliable credit assignment efficiently,

with an overview in Figure 2. We first prompt the off-the-shelf LLM-as-GenPRM to obtain the judgment for each step in a single pass. Then, we conduct credit assignment by setting asymmetric penalty to tokens in wrong steps. Finally, we use these credits to conduct RL training.

## 3.1 PROCESS CREDITS GENERATION VIA LLM-AS-GENPRM

To avoid the complex modeling and dependencies of existing fine-grained approaches, we propose to model the process credits in a deterministic manner. Our method focuses on the intrinsic correctness of each step, rather than its estimated future returns. To achieve this, we introduce an LLM as a GenPRM (LLM-as-GenPRM) to identify erroneous steps in the responses produced by the policy model rollouts during RL. A key advantage of this framework is its immediate applicability using existing open-source models without any domain-specific fine-tuning of the GenPRM itself. This not only validates the general reasoning capabilities of modern LLMs but also opens up a promising avenue for future work, where the GenPRM could be further fine-tuned on specific verification data for even greater accuracy and efficiency. We use the LLM to generate the rewards in a single pass for all steps efficiently. Given the training data $(q, a) \in \mathcal{D}$, we sample $n$ rollouts from the policy model $\pi_{\theta_{\text{old}}}$: $\{y_i\}_{i=1}^n \sim \pi_{\theta_{\text{old}}}(\cdot \mid q)$. Then for each rollout, we utilize LLM-as-GenPRM $\pi_{\text{LLM-as-GenPRM}}$ to generate $k$ critiques:

$$\{c_j\}_{j=1}^k \sim \pi_{(\text{LLM}-\text{as}-\text{GenPRM})}(\cdot \mid I, q, y_i),$$

where $I$ is the critic prompt for LLM-as-GenPRM. Next, we extract the step indexs of wrong steps from critiques: $\mathcal{S}_{i,j} = \text{ExtractIndices}(c_j), \forall j \in \{1, \ldots, k\}$, where $\mathcal{S}_{i,j}$ represents the set of wrong step indices identified by the $j$-th critique $c_j$ for the $i$-th rollout $y_i$. For each rollout $y_i$, this process yields $k$ sets of indices, denoted as $\{\mathcal{S}_{i,j}\}_{j=1}^k$.

These sets of indices are then aggregated through a *voting mechanism* to produce a final set $\mathcal{S}_i^*$ of erroneous step indices for rollout $y_i$. The voting process can employ various strategies, including:

- **Intersection Vote:** This conservative approach requires consistent agreement among all $k$ critiques. The final set contains only those step indices consistently identified as erroneous by all $k$ critiques: $\mathcal{S}_i^\cap = \bigcap_{j=1}^k \mathcal{S}_{i,j}$. This method maximizes precision and is robust to false positives, but may miss genuine errors not caught by all critiques, leading to lower recall.

- **Majority Vote:** A balanced strategy that declares a step as erroneous if it is flagged by at least half of the critiques. Let $\text{count}(s)$ be the number of times step index $s$ appears across all sets $\{\mathcal{S}_{i,j}\}_{j=1}^k$. The final set is $\mathcal{S}_i^{\text{maj}} = \{s \mid \text{count}(s) \geq k/2\}$. This offers a compromise between intersection and union.

A single sampled response carries high randomness from the model's stochastic decoding. By generating multiple critiques, we reduce the noise for robust and reliable identification of wrong steps. We here only present the implementation of the most representative voting mechanism for limited space. And the details of other voting mechanisms and discussion can be found in Appendix.

## 3.2 FINE-GRAINED CREDIT ASSIGNMENT

For a given rollout $y_i = (o_1, o_2, \ldots, o_{L_i})$, where $o_t$ is the $t$-th token and $L_i = |y_i|$ is the total number of tokens in the response, we aim to assign credit more precisely.

**Localizing Contributions with Token-level Rewards.** We initialize all the tokens in $y_i$ with the score of response-level reward $r_v$ from the rule-based verifier, scaled by a factor $W_{\text{whole}}$. Let $\mathcal{S}_i^*$ be the set of identified erroneous step indices for response $y_i$. We define $\mathcal{T}_{\text{err}}^i$ as the set of token indices belonging to these erroneous steps:

$$\mathcal{T}_{\text{err}}^i = \bigcup_{s \in \mathcal{S}_i^*} \{t \mid \text{token } o_t \text{ is part of step } s\}.$$

The token reward $R_t^i$ for token $o_t$ in response $y_i$ is then formulated as:

$$R_t^i = r_v \cdot W_{\text{whole}} - \mathbb{I}(t \in \mathcal{T}_{\text{err}}^i) \cdot W_{\text{process}}, \tag{1}$$

where $\mathbb{I}(t \in \mathcal{T}_{\text{err}}^i)$ is an indicator function that equals 1 if token $o_t$ is part of an erroneous step, and 0 otherwise. $W_{\text{process}}$ is the factor controlling the magnitude of the penalty. Equation 1 integrates

both process and outcome rewards into a single formulation. However, we empirically find that these two signals can conflict, creating a challenging optimization dynamic. Therefore, we conduct an analysis of their interaction and utilize *reward shaping* to balance their influence as below.

**The Shift from Outcome-Oriented Learning to Process-Oriented Learning.** The introduction of a penalty for process errors, as formulated in Equation 1, transforms the optimization *from a single-objective setting into a multi-objective one*. The model is no longer solely focused on the correctness of the final answer, but must also learn to focus on its intermediate steps. This necessitates a careful investigation into how these two reward signals interact and how their balance can be calibrated for optimal performance.

To analyze this interaction, we first consider the mode where process-based rewards dominate. Empirically, we observe that when the process penalty is too high, the response length and per-token reward increase, while final answer accuracy declines. We find that the model will overfit to a certain behavior pattern, optimizing for easily attainable process-level rewards (e.g., generating long, simple, but correct steps) at the expense of the ultimate goal. In contrast, a dominant outcome-based reward acts as a crucial corrective force, anchoring the policy back to the primary objective. We summarize our key insights about the mechanism of interaction of these two signals below, and the detailed discussion and validation can be found in the experiments and Appendix.

> **Observation 1**: *Learning from the process is a more challenging objective. In the early stages of optimization, accumulating correct process steps might not lead to a correct result, and even slow down the training.*

> **Observation 2**: *The process-oriented signal becomes critically important in the later stages of optimization. It helps to differentiate and prioritize samples with better reasoning among numerous responses that achieve the same outcome.*

These observations leads to our *reward shaping* design, which can be framed in the context of Multi-Objective Reinforcement Learning (MORL) (Mossalam et al., 2016; Wu et al., 2023). Our approach employs a *linear scalarization* of the two objectives. Based on the observations of the learning process, we adopt an *asymmetric configuration* ($W_{\text{whole}} > W_{\text{process}}$) in Equation 1 to construct the reward formulation. By setting $W_{\text{whole}} > W_{\text{process}}$, we ensure that any trajectory leading to a correct answer is *always* preferred over any trajectory leading to a wrong one, regardless of process errors. Within this primary constraint, the model is then secondarily encouraged to refine its reasoning process. For example, $W_{\text{whole}} = 2, W_{\text{process}} = 1$ yields a spectrum of distinct rewards that provides diverse, fine-grained feedback:

$$\text{Correct Answer, Correct Process: } R = 2 \cdot 1 - 1 \cdot 0 = +2$$
$$\text{Correct Answer, Incorrect Process: } R = 2 \cdot 1 - 1 \cdot 1 = +1$$
$$\text{Incorrect Answer, Correct Process: } R = 2 \cdot 0 - 1 \cdot 0 = \phantom{+}0$$
$$\text{Incorrect Answer, Incorrect Process: } R = 2 \cdot 0 - 1 \cdot 1 = -1$$

This structured hierarchy provides a robust and interpretable learning signal. More detailed analysis of the weights $W_{\text{whole}}$ and $W_{\text{process}}$ is provided in our ablation study.

**Group-Relative Per-Token Normalization.** After obtaining the per-token reward $R_t^i$, the advantage $\hat{A}_t^i$ for token $o_t$ in rollout $y_i$ is calculated by normalizing $R_{i,t}$ using the mean and standard deviation of all per-token rewards across a batch of $n$ rollouts. Let $\{(y_j, \{R_{t'}^j\}_{t'=1}^{L_j})\}_{j=1}^n$ be the set of rollouts and the per-token rewards in the batch. The fine-grained advantage estimate is:

$$\hat{A}_t^i = \frac{R_t^i - \text{mean}(\{R_{t'}^j\}_{j=1,\ldots,n,\ t'=1,\ldots,L_j})}{\text{std}(\{R_{t'}^j\}_{j=1,\ldots,n,\ t'=1,\ldots,L_j})}. \tag{2}$$

This formulation ensures that *the advantage estimates directly reflect the reliable, fine-grained and token-level feedback*, guiding the policy update to specifically suppress the generation of erroneous segments while appropriately rewarding correct ones. Then, we derive the final objective function below and subsequently use this objective function to update the policy model:

$$\mathcal{J}_{\text{CAPO}}(\theta) = \mathbb{E}_{q \sim Q,\ \{y_i\}_{i=1}^n \sim \pi_{\theta_{\text{old}}}(o|q)}$$
$$\frac{1}{n} \sum_{i=1}^n \frac{1}{L_i} \sum_{t=1}^{L_i} \min \left\{ \frac{\pi_\theta(o_{i,t}|q, o_{i,<t})}{\pi_{\theta_{\text{old}}}(o_{i,t}|q, o_{i,<t})} \hat{A}_t^i, \text{clip}\left(\frac{\pi_\theta(o_{i,t}|q, o_{i,<t})}{\pi_{\theta_{\text{old}}}(o_{i,t}|q, o_{i,<t})}, 1-\epsilon, 1+\epsilon\right) \hat{A}_t^i \right\} - \beta \mathbb{D}_{KL}\left[\pi_\theta \| \pi_{\text{ref}}\right].$$

## 4 EXPERIMENTS

### 4.1 EXPERIMENTS SETUPS

**Implementation Details.** To validate the effect of diversity generalization of our method, We conduct experiments on Llama-3-1B/3B-Instruct (Dubey et al., 2024), Qwen2.5-1.5B/7B-Base (Team, 2024) for mathematical reasoning tasks. For Llama-3-1B-Instruct and Llama-3-3B-Instruct, we directly perform RL on the MATH dataset (Hendrycks et al., 2021) with 7.5k samples and the more challenging DAPO-Math dataset (Yu et al., 2025) with 9.8k samples, respectively. For Qwen2.5-1.5B/7B-Base, We first conduct SFT using NuminaMath-CoT (LI et al., 2024) with 860K samples. Then for 1.5B model, we conduct RL using MATH dataset. And for 7B model, we train the model using DAPO-Math dataset. As for the generative reward model, we utilize Qwen2.5-72B-Instruct as LLM-as-GenPRM for Qwen series experiments and Llama-3-70B-Instruct for Llama series experiments without fine-tuning for the step-wise verification task. For the voting mechanisms, in the main experiments, we use N=4 critiques with different voting mechanisms to balance the trade-off between efficiency and performance. All the hyperparameters of RL and SFT training could be found in Table 11. We also conduct experiments using the 8k difficulty-stratified dataset following Zero RL setting in Zeng et al. (2025) without cold start SFT and **extend to the larger base model like Qwen2.5-14B**. The results are shown in Table 5, 6 and 8. **In this series of experiments, to further verify the weak dependency of our method on the scale and capability of the GenPRM, we employ Qwen2.5-14B/32B-Instruct as the GenPRM.** We use \n \n as a delimiter following the widely accepted convention in the Process Reward Model literature (Zhang et al., 2025b; Yang et al., 2024; Zheng et al., 2025). All the experiments are conducted on 96 NVIDIA A100 40G GPUs.

**Evaluation Setups.** We evaluate the performance on four mathematical reasoning benchmarks: MATH (Hendrycks et al., 2021), Olympiadbench (He et al., 2024), AMC2023 (Mathematical Association of America, 2023), AIME2024 (Mathematical Association of America, 2024). We also include evaluation on three out-of-distribution benchmarks ARC-c (Clark et al., 2018), MMLU-Pro (Wang et al., 2024), GPQA-diamond (Rein et al., 2024), foucusing on open-domain, scientific and general reasoning. To avoid contamination, we follow (Yan et al., 2025) to shuffle the multiple-choice options. During evaluation, we use greedy decoding and the final results (pass@1) are averaged over three runs.

**Baselines.** We compared our methods with Supervised Finetuning (SFT), GRPO (Guo et al., 2025) with rule-based verifier, with generative outcome reward models (GenORM) as the verifier, Process Reward Models based approaches (Cheng et al., 2025) and other RLVR approaches like RLOO Ahmadian et al. (2024). We train the models to convergence and report the best performance. Since our method can adapt on other RLVR methods like DAPO (Yu et al., 2025), VAPO (Yue et al., 2025), Reinforce++ (Hu, 2025) and so on, we mainly equip our methods upon GRPO to evaluate the effectiveness of our method.

Table 2: Comparison of results (Pass@1) across different methods on Llama-3-3B and Qwen2.5-7B for mathematical (In-Distribution) and general (Out-of-Distribution) reasoning tasks. N=4 critiques for CAPO with voting.

| Method | General Reasoning Tasks (OOD) | | | | Math Reasoning Tasks (ID) | | | | | All Mean |
|---|---|---|---|---|---|---|---|---|---|---|
| | GPQA Diamond | ARC_c | MMLU Pro | OOD Mean | MATH 500 | Olympiad Bench | AIME24 | AMC23 | Math Mean | |
| *Llama-3-3B* | | | | | | | | | | |
| SFT | 19.7 | 56.1 | 21.0 | 32.3 | 43.2 | 12.7 | 6.7 | 10.8 | 18.4 | 24.3 |
| GRPO-Rule | 19.9 | 61.2 | **32.4** | 37.8 | 44.6 | 15.0 | **10.0** | 16.9 | 21.6 | 28.6 |
| GRPO-GenORM | 22.2 | 62.2 | 31.8 | 38.7 | **46.6** | 14.8 | 3.3 | 20.0 | 21.2 | 28.5 |
| CAPO-Greedy | **24.2** | 61.9 | 31.7 | **39.3** | 45.6 | 14.1 | 3.3 | 19.3 | 20.6 | 28.6 |
| CAPO-Intersection | 23.2 | **63.2** | 31.5 | **39.3** | 44.3 | 15.6 | 7.8 | 20.3 | 22.0 | 29.4 |
| CAPO-MajorVote | 20.9 | 62.4 | 31.5 | 38.3 | 44.6 | **16.7** | 6.7 | 26.8 | **23.7** (+2.1%) | **30.4** (+1.8%) |
| *Qwen-2.5-7B* | | | | | | | | | | |
| SFT | **23.7** | 67.7 | 32.5 | 41.3 | 60.7 | 28.1 | 3.3 | 33.0 | 31.3 | 34.7 |
| GRPO-Rule | 22.4 | 68.1 | 32.7 | 41.0 | 59.9 | 27.2 | 3.6 | 34.1 | 31.2 | 34.5 |
| GRPO-GenORM | 23.6 | 68.0 | **32.9** | **41.5** | 59.2 | 27.6 | 4.0 | 33.4 | 31.1 | 34.5 |
| CAPO-Greedy | 21.5 | 68.2 | 32.9 | 40.9 | 62.4 | **31.3** | 9.7 | 34.2 | 34.4 | **36.8** |
| CAPO-Intersection | 22.2 | **68.3** | 32.9 | 41.2 | 62.5 | 30.5 | 7.1 | **34.6** | 33.7 | 36.3 |
| CAPO-MajorVote | 22.7 | 67.0 | 32.8 | 40.8 | **63.4** | 31.0 | 10.8 | 34.1 | **34.8** (+3.5%) | **37.0** (+2.3%) |

### 4.2 MAIN RESULTS

As shown in Table 3 and Table 2, our proposed method, CAPO, consistently outperforms various baselines across various backbone models and scales. The detailed statistical results like variance

Table 3: Comparison of results (Pass@1) across different methods on Llama-3-1B and Qwen2.5-1.5B for mathematical (In-Distribution) and general (Out-of-Distribution) reasoning tasks. N=4 critiques for CAPO with voting.

| Method | General Reasoning Tasks (OOD) | | | | Math Reasoning Tasks (ID) | | | | All Mean |
|---|---|---|---|---|---|---|---|---|---|
| | GPQA Diamond | arc_c | MMLU Pro | OOD Mean | MATH 500 | Olympiad Bench | AMC23 | Math Mean | |
| *Llama-3-1B* | | | | | | | | | |
| SFT | 13.1 | 19.1 | **10.1** | 14.1 | 22.0 | 6.7 | 8.8 | 12.5 | 13.3 |
| GRPO-Rule | **28.8** | 21.6 | 4.5 | 18.3 | 27.1 | **7.0** | 6.2 | 13.4 | 15.9 |
| GRPO-GenORM | 25.6 | 22.0 | 4.4 | 17.4 | 27.2 | 6.3 | 7.6 | 13.7 | 15.5 |
| CAPO-Greedy | 25.6 | 28.5 | 5.7 | 19.9 | 26.3 | 5.7 | 9.8 | 13.9 | 16.9 |
| CAPO-Intersection | 25.3 | **34.0** | 6.2 | 21.8 | **28.7** | 6.5 | **12.0** | **15.7** (+2.0%) | **18.8** (+2.9%) |
| CAPO-MajorVote | 21.9 | 32.3 | 5.7 | 20.0 | 26.4 | 6.0 | 9.2 | 13.9 | 16.9 |
| *Qwen-2.5-1.5B* | | | | | | | | | |
| SFT | 16.0 | **61.1** | 20.9 | 32.7 | 47.5 | 16.0 | **26.1** | 29.9 | 31.3 |
| GRPO-Rule | 20.4 | 60.6 | 20.7 | 33.9 | 48.9 | 17.0 | 20.4 | 28.8 | 31.3 |
| GRPO-GenORM | 18.2 | 60.8 | 20.9 | 33.3 | 49.2 | 17.4 | 20.2 | 28.9 | 31.1 |
| CAPO-Greedy | 14.8 | 59.8 | **21.9** | 32.2 | 49.3 | 18.1 | 22.4 | 29.9 | 31.1 |
| CAPO-Intersection | **20.5** | 60.0 | 21.8 | 34.1 | 48.7 | **18.8** | 24.0 | **30.5** (+0.6%) | **32.3** (+1.0%) |
| CAPO-MajorVote | 19.9 | 60.4 | 21.6 | 34.0 | **49.7** | 18.6 | 22.0 | 30.1 | 32.0 |

Table 4: Comparison of **Step Error Ratio (SER)** across different methods using Llama3.2-1B-Instruct, Qwen2.5-Math-1.5B, and Qwen2.5-Math-7B. Lower is better. This table is newly added.

| Method | General Reasoning Tasks (OOD) | | | | Math Reasoning Tasks (ID) | | | | | |
|---|---|---|---|---|---|---|---|---|---|---|
| | GPQA | ARC-C | MMLU Pro | OOD Mean | MATH | Olympiad Bench | Minerva | AIME | AMC | Math Mean |
| *Llama3.2-1B-Instruct* | | | | | | | | | | |
| GRPO | 55.65 | 25.22 | 47.33 | 42.73 | 40.31 | 56.49 | 56.56 | 67.29 | 52.06 | 54.54 |
| CAPO | **49.02** | **23.06** | **45.73** | **39.27** | **37.95** | **51.72** | **55.54** | **51.10** | **49.01** | **49.06** |
| *Qwen2.5-Math-1.5B* | | | | | | | | | | |
| GRPO | 37.74 | 23.66 | 33.43 | 31.61 | 10.93 | 22.62 | 19.96 | 27.62 | 19.64 | 20.15 |
| CAPO | **36.31** | **20.60** | **27.41** | **28.11** | **7.45** | **17.79** | **15.52** | **26.01** | **13.01** | **15.96** |
| *Qwen2.5-Math-7B* | | | | | | | | | | |
| GRPO | 29.55 | 12.94 | 30.86 | 24.45 | **4.96** | 15.24 | 15.10 | 32.83 | 12.31 | 16.09 |
| CAPO | **24.76** | **7.44** | **20.29** | **17.49** | 5.98 | **13.93** | **10.72** | **20.16** | **10.46** | **12.25** |

can be found in Appendix. For Qwen-2.5-7B, CAPO delivers a remarkable +3.5 point gain on math benchmarks among all the baselines and drives an overall average improvement of +2.5 points in all the benchmarks. For Qwen-2.5-1.5B, CAPO achieves +1.7 points lead in math reasoning tasks compared with GRPO without credit assignment. On the Llama series, CAPO also shows clear improvements. For Llama-3-1B, we achieve a +3.5 points gain on OOD tasks among all the baselines, also leading to a strong +2.9 points overall improvement. On the larger Llama-3-3B, CAPO particularly excels in math reasoning, establishing a solid +2.1 points lead over the best baseline and +1.8 points in all benchmarks.

A key observation is that across all backbones, CAPO and its variants consistently outperform GRPO with varia which forgoes explicit credit assignment. This indicates that the integration of our credit assignment mechanism is the primary driver of these improvements, with benefits realized regardless of the specific voting mechanism employed. Crucially, these gains are not confined to mathematical reasoning; the approach also enhances the model's generalization capabilities on a wide range of common reasoning tasks.

**Robustness with Smaller GenPRMs** As shown in Tables 8, 6 and 5, CAPO consistently outperforms GRPO, RLOO, and the PRM-based method even when **using these smaller, more efficient GenPRMs**. This demonstrates that CAPO does not require prohibitive compute to be effective; the fine-grained signal provided by a moderately sized judge **(e.g., 14B)** is sufficient. This indicated that our method can generalize to smaller judges and across various LLMs.

**CAPO Indeed Improve the Reasoning Quality** To rigorously quantify the quality of the reasoning process, we define two metrics: **Step Error Ratio (SER)**: The percentage of reasoning steps identified as incorrect and **Token Error Ratio(TER)**: The percentage of total tokens belonging to incorrect steps. Let dataset $\mathcal{D}$ contain $N$ generated responses. For the $i$-th response $Y_i$, let it consist of $K_i$ steps $(s_{i,1}, \ldots, s_{i,K_i})$, where $|s_{i,k}|$ denotes the token count of step $s_{i,k}$, and $\mathbb{I}(s_{i,k})$ is the

Table 5: Performance comparison across different methods using Qwen2.5-14B on Zero RL setting without SFT. CAPO uses Qwen2.5-32B-Instruct as a judge. This table is newly added.

| Method | General Reasoning Tasks (OOD) | | | | Math Reasoning Tasks (ID) | | | | | | | All Mean |
|---|---|---|---|---|---|---|---|---|---|---|---|---|
| | GPQA | ARC-C | MMLU Pro | OOD Mean | MATH | Olympiad Bench | Minerva | AIME24 | AMC | AIME25 | Math Mean | |
| Base | 25.76 | 76.54 | 54.89 | 52.39 | 61.60 | 29.19 | 25.37 | 10.00 | 33.85 | 5.21 | 27.53 | 35.82 |
| GRPO | 30.30 | 86.18 | 61.96 | 59.48 | **81.40** | **44.59** | 38.60 | 16.67 | 50.45 | 5.63 | 39.56 | 46.20 |
| CAPO | **30.81** | **90.19** | **62.06** | **61.02** | 80.00 | 42.96 | **41.91** | **22.81** | **52.97** | **13.44** | **42.35** | **48.57** |

Table 6: Performance comparison across different methods using Qwen2.5-Math-1.5B and Qwen2.5-Math-7B on Zero RL setting without SFT. CAPO uses Qwen2.5-14B-Instruct as a judge. This table is newly added.

| Method | General Reasoning Tasks (OOD) | | | | Math Reasoning Tasks (ID) | | | | | | All Mean |
|---|---|---|---|---|---|---|---|---|---|---|---|
| | GPQA | ARC-C | MMLU Pro | OOD Mean | MATH | Olympiad Bench | Minerva | AIME | AMC | Math Mean | |
| *Qwen2.5-Math-1.5B* | | | | | | | | | | | |
| Base | 5.05 | 18.20 | 10.30 | 11.18 | 63.13 | 29.43 | 12.25 | 9.97 | 41.57 | 31.27 | 23.74 |
| GRPO | **14.65** | **50.09** | 23.48 | **29.40** | 76.40 | 38.32 | 31.62 | 6.67 | 42.31 | 39.06 | 35.44 |
| RLOO | 13.47 | 39.11 | 21.60 | 24.73 | 75.60 | 35.70 | 29.66 | 6.67 | 49.75 | 39.48 | 33.94 |
| PRM | 10.61 | 45.88 | 22.99 | 26.49 | 76.00 | 38.81 | **32.35** | 12.40 | 43.30 | 40.57 | 35.29 |
| CAPO | 13.13 | 43.34 | **24.28** | 26.92 | **76.80** | **40.20** | 32.23 | 18.19 | 44.73 | 42.43 | **36.61** |
| CAPO_GenRM32B | 13.64 | 41.81 | 20.46 | 25.30 | 74.70 | 38.00 | 28.86 | **23.18** | **49.85** | **42.92** | 36.31 |
| *Qwen2.5-Math-7B* | | | | | | | | | | | |
| Base | 18.18 | 30.32 | 16.71 | 21.74 | 64.73 | 19.80 | 9.07 | 11.35 | 42.28 | 29.45 | 26.56 |
| GRPO | **25.76** | 64.11 | **44.62** | **44.83** | 81.13 | 44.10 | 36.76 | 30.00 | 56.78 | 49.75 | 47.91 |
| RLOO | 22.22 | 42.43 | 41.84 | 35.50 | 79.33 | 41.73 | 33.95 | **40.52** | 59.46 | 51.00 | 45.19 |
| PRM | 20.03 | **70.73** | 42.08 | 44.28 | 81.60 | 43.11 | **38.36** | 24.34 | **61.02** | 49.69 | 47.66 |
| CAPO | 23.89 | 65.85 | 43.20 | 44.31 | **81.67** | **45.14** | 38.11 | 35.00 | 59.10 | **51.80** | **48.99** |

error indicator (1 if incorrect, 0 otherwise). We calculate the metrics by averaging the error rates per response: $\text{SER} = \frac{1}{N}\sum_{i=1}^{N}\left(\frac{\sum_{k=1}^{K_i} \mathbb{I}(s_{i,k})}{K_i}\right)$, and $\text{TER} = \frac{1}{N}\sum_{i=1}^{N}\left(\frac{\sum_{k=1}^{K_i} \mathbb{I}(s_{i,k})\cdot|s_{i,k}|}{\sum_{k=1}^{K_i}|s_{i,k}|}\right)$. As shown in the Table 4 and Table 10, CAPO demonstrates a **significant reduction in process errors compared to GRPO** across both In-Domain (Math) and Out-Of-Domain benchmarks and achieves an average reduce up to 7%. The lower Error Token Ratio confirms that CAPO generates reasoning chains with fewer logic or calculation errors. These findings directly validate the motivation of CAPO.

**Robustness to the Selection of LLM-as-GenPRM**  We also highlight the robustness of our framework with different LLMs as GenPRMs. We utilized off-the-shelf, open-source LLMs Qwen2.5-14B/32B/72B-Instruct and Llama-3-70B-Instruct as the GenPRM without any further fine-tuning for the step-wise verification task. The fact that our method proved effective in both settings demonstrates low sensitivity to the LLMs choice, underscoring the versatility, generality, and broad applicability of our method. This successful application of different off-the-shelf models without any GenPRM-specific fine-tuning is a strong testament to CAPO's simplicity and generality. It demonstrates that our method is not reliant on a specific, proprietary verifier, but can effectively leverage the general reasoning abilities of various publicly available LLMs. This significantly enhances the method's practical utility and makes it easily adoptable for the broader research community to build upon.

**Analysis of LLM-as-GenPRM Scaling**  We performed an analysis of the impact of the number of generated critiques ($N$) from LLM-as-GenPRM on final performance. As shown in Figrue 3b and Figure 3a, we can observe a consistent scaling trend across both math and general reasoning tasks. And when $N$ reaches to 8, the overall performance is best across math and OOD reasoning tasks. The results show that more critiques from LLM-as-GenPRM can help to obtain a more accurate judgment of incorrect steps and therefore provide a more robust signal for credit assignment, leading to better final performance. With a small $N$, the judgment can be easily skewed by a few noisy. By increasing $N$, we can use the voting mechanism to form a consensus, averaging out these outliers to get a more accurate judgment and therefore leading to more accurate credit assignment.

**Ablation Study on Reward Formulation**  Our key insight lies in the C/P balance: an *asymmetric* setup where C > P is crucial for creating a clear learning hierarchy. Without this balance, the reward

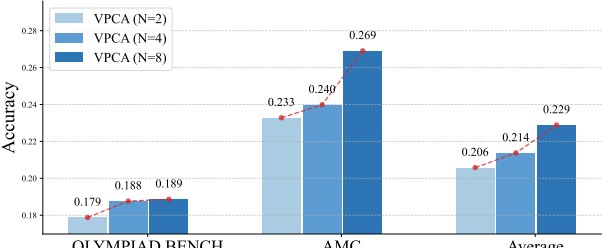

(a) Effect of the number of critiques ($N$) on CAPO performance.

(b) Performance on OOD benchmarks for varying $N$.

| Model | GPQA Dia. | ARC_c | OOD Mean |
|---|---|---|---|
| CAPO ($N$=2) | 17.5 | 60.4 | 33.2 |
| CAPO ($N$=4) | 20.5 | 60.0 | 34.1 |
| CAPO ($N$=8) | **22.1** | **61.1** | **35.0** |

Figure 3: Performance analysis of CAPO with a varying number of critiques ($N \in \{2, 4, 8\}$) generated by LLM-as-GenPRM on Qwen2.5-1.5B CAPO-Intersection. (a) Performance trend on the math dataset. (b) Detailed performance on OOD benchmarks.

signal becomes ambiguous. For instance, a *symmetric* (C=P) setting can assign a zero reward to both a "Answer Correct, Process Incorrect" ($2 \cdot 1 - 2 = 0$) and "Answer Incorrect, Process Correct" ($2 \cdot 0 = 0$), confusing the model. An extreme process-focus (C<P) breaks the hierarchy entirely, penalizing a single mistake in a correct answer more than a completely failed attempt. The results in Table 7 are consistent with this analysis. The asymmetric (C=2, P=1) setup clearly outperforms all others. The setting (C=2, P=0.1), with its weak process penalty, performs less well, *demonstrating the necessity of a meaningful process signal*. The poor performance of the symmetric(C=2, P=2) and extreme process-focused (C=2, P=5) settings confirms that a poorly calibrated reward signal (imbalanced or ambiguous) can misguides the model.

**Discussion of Per-Token Rewards and Response Length** Figure 1b shows that CAPO consistently generates longer responses than GRPO, suggesting the model is learning more *elaborate and effective reasoning strategies* by exploring solution paths more thoroughly. Critically, this increased length is coupled with higher quality, as the average token-level reward also rises. This trend is significant. Longer responses could simply introduce more errors, but the parallel improvement in rewards demonstrates that *CAPO is not merely rewarding verbosity*. Instead, it teaches the model to maintain logical coherence and accuracy even within these extended reasoning trajectories.

## 5 ADDITIONAL EXPERIMENTS

Due to the limited space, we provide a detailed discussion and limitation about CAPO in Appendix C and additional experiments in Appendix E about the detailed comparison of GPU hours of CAPO and GRPO, **discussion of process-reward and outcome-reward, discussion of different voting mechanisms, the impact of incorporation of ground truth solution, the discussion and comparisons between GenORM and GenPRM, the case study of different methods, and the case study of internal variations in rollout responses.**

## 6 RELATED WORK

Our work is related to research on enhancing LLM reasoning with RL, credit assignment in RL and generative reward modeling. Due to limited space, we offer a detailed discussion in Appendix B.

## 7 CONCLUSION

In this work, we first identify the challenge to efficiently conduct precise credit assignment in RLVR. Then, we propose an elegantly simple, and highly generalizable method–Credit Assignment Policy Optimization (CAPO) to address the challenge. Our approach bypasses the need for complex auxiliary models by using an off-the-shelf, general-purpose LLM as a generative process reward model to generate all step-wise critique in a single pass to obtain reliable credits. We further enhance reward robustness by integrating multiple critiques through a voting mechanism. We also design an elaborate reward formulation to balance the outcome and process reward signals. Extensive experiments show that CAPO outperforms existing RLVR methods without accurate credit assignment on challenging math and general reasoning benchmarks. Further analysis shows that CAPO can guide the model to foster the learning of correct reasoning pathways, promoting more effective exploration. Crucially, the simplicity and generality of our framework make it a practical and accessible tool, fostering reproducibility and paving the way for future innovations in LLM reasoning.

## 8 ETHICS STATEMENT

The research presented in this paper is focused on fundamental advancements in the mathematical and logical reasoning capabilities of large language models. Our goal is to improve the reliability and correctness of the models' internal thought processes, contributing to the development of more robust and trustworthy AI systems. The scope of our work is confined to well-defined, objective reasoning tasks using publicly available, standard academic benchmarks. These datasets do not contain sensitive personal information, and our research does not involve human subjects. Consequently, our work does not directly engage with issues such as societal bias, fairness, or the generation of harmful content, as the problem domain is primarily mathematical and abstract. We believe that enhancing the foundational reasoning abilities of AI is a crucial and beneficial step for the field. We have conducted our research in accordance with the ICLR Code of Ethics.

## 9 REPRODUCIBILITY STATEMENT

We are committed to ensuring the reproducibility of our research. Our implementation of CAPO, including the scripts and code, is provided in the supplementary material and `https://anonymous.4open.science/r/CAPO_Repo-45F1/`. The core methodology of our proposed approach is detailed in Section 3. Key implementation details, including the specific prompts used for our LLM-as-GenPRM, are available in Appendix F. Section 4.1 describes our complete experimental setup, including the datasets, model backbones, and evaluation benchmarks used. A comprehensive list of all hyperparameters for both SFT and RL training stages is provided in Table 11 (Appendix D). Further in-depth analyses of our design choices, such as the reward shaping strategy (Appendix E) and the various voting mechanisms (Appendix E), are also included in the appendix to support our claims and facilitate verification. Together, these resources provide a clear and complete basis for replicating our results.

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

# APPENDIX

## A    STATEMENT ON LLM USAGE

In the preparation of our paper, Large Language Models (LLMs) were utilized in a limited and supportive capacity. First, LLMs served as a writing aid to enhance the grammatical correctness and clarity of the text. Suggestions for improving sentence structure and flow were considered, but the authors retained full authority over the final composition, ensuring that all arguments and scientific nuances accurately reflect our original intent. Second, LLMs were employed to assist in formatting citations and references according to the journal's specific style guidelines, which helped streamline the manuscript preparation process. Importantly, LLMs were not involved in any core aspect of the research process. This includes the formulation of the research question, the design of the methodology, the collection and analysis of data, and the interpretation of results. All intellectual contributions and substantive findings presented herein are solely the work of the authors.

## B    RELATED WORK

**Improving LLM Reasoning Capabilities Using RL** Reinforcement learning has been widely applied to enhance LLMs' reasoning capabilities, but many challenges remain-such as optimization difficulties introduced by long reasoning (Yuan et al., 2025), how to better balance and improve the exploration–exploitation trade-off in the RL process (Yue et al., 2025), clipping-ratio issues (Yu et al., 2025), entropy decrease during RL (Wang et al., 2025), and so on. A range of algorithms aimed at improving RL have since emerged in the community, including GRPO (Shao et al., 2024), Reinforce++ (Hu, 2025), VC-PPO (Yuan et al., 2025), VAPO (Yue et al., 2025), DAPO (Yu et al., 2025). These works enhance existing RL algorithms from an algorithmic-optimization perspective, and we address the problem of imprecise or non-verifiable credit assignment in current approaches.

**Credit Assignment** *Credit assignment problem* (Sutton et al., 1998; Arumugam et al., 2021; Zhou et al., 2020) is a fundamental RL challenge. A key challenge is how to conduct accurate credit assignment in RL finetuning of LLMs. Recent literature has focused on resolve this through several key strategies. One approach is reward signal densification. This involves decomposing sequence-level rewards to token-level feedback, leveraging reward model attention mechanisms (Chan et al., 2024) or game-theoretic principles like Shapley values (Cao et al., 2025) for fair attribution. However, these methods are non-verifiable relying a reward model or other unverifiable reward signals, posing potential reward-hacking problems. We call the method using process reward model (PRM) to assign credit as PRM-based methods. Although both reward-densification-based and PRM-based methods provide credit assignment, they suffer from significant limitations. Due to factors such as unreliable labeled data (Zhang et al., 2025c) and mismatches in the completion model (Setlur et al., 2024), PRMs are often inaccurate, which means that they can't offer reliable or accurate rewards in online-rl and ultimately leads to the training of a suboptimal policy. PRM-based methods also need multiple inference calls to get the process rewards of each step which is costly especially for the response with long reasoning. Reward-densification-based methods (Chan et al., 2024; Cao et al., 2025) did not need the multiple inference calls and did not need high-quality data. However, these methods still suffer from the problem of inaccurate estimations of the credits and they can't offer reliable process rewards. VinePPO (Kazemnejad et al., 2024) proposed to replace unreliable value function learning with value-free Monte Carlo estimates, but with huge computational costs.

**Generative Reward Model** Recently, several works focused on using capabilities of language modeling in LLMs to improve reward modeling (Mahan et al., 2024; Zhang et al., 2024) and further scale it with COT reasoning (Zhang et al., 2024; Liu et al., 2025). And this technique is similar to previous concept "LLM as a judge" (Zheng et al., 2023) using LLMs to judge the response. Zhang et al. (2024) propose Generative Verifiers, which recast reward modeling as a next-token prediction task, thereby enabling integrated chain-of-thought reasoning and leveraging test-time majority voting for more robust ranking. Recently, Liu et al. (2025) introduced Generative Reward Modeling for flexible, inference-time scalable reward estimation and propose a more sophisticated technique Self-Principled Critique Tuning (SPCT) to further scale the performance of GenRM.

Table 7: Performance comparison of CAPO with different reward configurations of $W_{\mathrm{whole}}$(C) and $W_{\mathrm{process}}$(P) on Qwen2.5-1.5B. The asymmetric setting leads to the best performance.

| Method | General Reasoning Tasks (OOD) | | | | Math Reasoning Tasks (ID) | | | | All Mean |
|---|---|---|---|---|---|---|---|---|---|
| | GPQA Diamond | ARC_c | MMLU Pro | OOD Mean | MATH 500 | Olympiad Bench | AMC23 | Math Mean | |
| CAPO (C=2, P=0.1) | 18.9 | 59.3 | **22.0** | 33.4 | 47.8 | 17.6 | 22.5 | 25.4 | 28.8 |
| CAPO (C=2, P=5) | 15.7 | **62.2** | 21.1 | 33.0 | 46.8 | 15.1 | 16.6 | 23.2 | 27.4 |
| CAPO (C=2, P=2) | 16.2 | 61.1 | 21.3 | 32.8 | **51.1** | 17.5 | 19.9 | 25.2 | 28.5 |
| CAPO (C=2, P=1) | **20.7** | 59.9 | 21.6 | **34.1** | 47.6 | **19.1** | 23.9 | 25.9 | **29.4** |

Table 8: Performance Comparison across different methods using Llama3.2-1B-Instruct on Zero RL setting without SFT. CAPO uses Qwen2.5-14B-Instruct as a judge. This table is newly added.

| Method | General Reasoning Tasks (OOD) | | | | Math Reasoning Tasks (ID) | | | | | All Mean |
|---|---|---|---|---|---|---|---|---|---|---|
| | GPQA | ARC-C | MMLU Pro | OOD Mean | MATH | Olympiad Bench | AMC | GSM8K | Math Mean | |
| Base | 18.69 | 6.91 | 1.30 | 8.96 | 22.60 | 4.89 | 5.65 | 20.62 | 13.44 | 11.52 |
| GRPO | 20.54 | 51.68 | 17.16 | 29.79 | 31.00 | **8.44** | 11.23 | 51.28 | 25.49 | 27.33 |
| RLOO | **24.41** | 32.99 | 16.39 | 24.60 | 30.13 | 5.63 | 14.48 | 48.42 | 24.67 | 24.64 |
| PRM | 14.98 | 52.59 | 16.58 | 28.05 | 31.87 | 7.01 | 12.58 | 51.07 | 25.63 | 26.67 |
| CAPO | 17.60 | **54.78** | **17.56** | **29.98** | **33.13** | 8.30 | **17.41** | **51.55** | **27.60** | **28.62** |

Table 9: Performance comparison of different steps using Qwen2.5-Math-7B. CAPO uses Qwen2.5-14B-Instruct as a judge. The table is newly added.

| Method | Step | General Reasoning Tasks (OOD) | | | | Math Reasoning Tasks (ID) | | | | | Math Mean | All Mean |
|---|---|---|---|---|---|---|---|---|---|---|---|---|
| | | GPQA | ARC-C | MMLU Pro | OOD Mean | MATH | Olympiad Bench | Minerva | AIME | AMC | | |
| Base | – | 18.18 | 30.32 | 16.71 | 21.74 | 64.73 | 19.80 | 9.07 | 11.35 | 42.28 | 29.45 | 26.56 |
| GRPO | 60 | 23.23 | 59.44 | 43.69 | 42.12 | 80.53 | 41.53 | 38.73 | 26.67 | **61.38** | 49.77 | 46.90 |
| | 90 | 22.90 | 65.22 | **44.70** | 44.27 | 79.33 | 40.89 | 38.85 | 28.96 | 50.62 | 47.73 | 46.43 |
| | 120 | 24.75 | 60.75 | 44.69 | 43.40 | 81.93 | 43.56 | 34.56 | 24.17 | 60.17 | 48.88 | 46.82 |
| | 176 | **25.76** | 64.11 | 44.62 | **44.83** | 81.13 | 44.10 | 36.76 | 30.00 | 56.78 | 49.75 | 47.91 |
| CAPO | 60 | 22.89 | 58.06 | 43.94 | 41.63 | 81.33 | 43.17 | 38.03 | 28.11 | 60.49 | 50.23 | 47.00 |
| | 90 | 20.01 | 60.89 | 43.94 | 41.61 | **82.27** | 43.11 | **38.97** | 30.00 | 60.99 | 51.07 | 47.52 |
| | 120 | 23.89 | 65.85 | 43.20 | 44.31 | 81.67 | **45.14** | 38.11 | **35.00** | 59.10 | **51.80** | **48.99** |
| | 176 | 23.74 | **69.23** | 39.57 | 44.18 | 80.33 | 40.30 | 37.38 | 26.56 | 50.94 | 47.10 | 46.01 |

## C    DISCUSSION AND LIMITATIONS

**The Reliability and the Capability of the LLM-as-a-GenPRM**    A key design choice in CAPO is the use of a powerful, general-purpose LLM as the GenPRM. The success of this approach hinges on the verifier's ability to accurately identify reasoning errors. One might question the reliability of such a model. However, we argue this design is robust for two primary reasons. Firstly, we strategically employ a GenPRM that is more capable (but not necessarily extremely more capable) than the policy model being trained (e.g., using a 14B+ model to guide a 1B-7B model). This capability gap creates a natural "teacher-student" dynamic, where the stronger model's reasoning serves as a reliable guide for the smaller one (Yan et al., 2025; Dong et al., 2023). Secondly, and critically, our method provides the GenPRM with the ground-truth answer as a reference. This transforms the task from abstract, ungrounded verification to a more constrained problem of error localization. The GenPRM is not asked "is this reasoning correct in a vacuum?", but rather "given the correct final answer, can you trace back and identify the step where the model's logic diverged?". This grounded context dramatically simplifies the verification task and enhances its reliability, as the model can cross-reference the intermediate steps with the known correct outcome.

**Computational Considerations in Online RL**    CAPO integrates an external LLM call into the online RL loop, which naturally introduces computational overhead compared to methods relying solely on a simple rule-based verifier. But we also offer some key insights to demonstrate the relatively low or, in practice, lower cost introduced by CAPO and the efficiency of CAPO via faster convergence and higher performance upper bound.

Table 10: Comparison of **Token Error Ratio (TER)** across different methods using Llama3.2-1B-Instruct, Qwen2.5-Math-1.5B, and Qwen2.5-Math-7B. Lower is better.

| Method | General Reasoning Tasks (OOD) | | | | Math Reasoning Tasks (ID) | | | | | |
|---|---|---|---|---|---|---|---|---|---|---|
| | GPQA | ARC-C | MMLU Pro | OOD Mean | MATH | Olympiad Bench | Minerva | AIME | AMC | Math Mean |
| *Llama3.2-1B-Instruct* | | | | | | | | | | |
| GRPO | 59.41 | 22.80 | 45.30 | 42.50 | 38.40 | 56.63 | 57.73 | 67.66 | 52.83 | 54.65 |
| CAPO | **49.43** | **21.19** | **41.59** | **37.40** | **34.71** | **49.94** | **54.46** | **50.41** | **48.76** | **47.66** |
| *Qwen2.5-Math-1.5B* | | | | | | | | | | |
| GRPO | 48.01 | 23.74 | 33.03 | 34.93 | 11.12 | 26.34 | 24.94 | 41.11 | 23.99 | 25.50 |
| CAPO | **44.55** | **20.16** | **28.32** | **31.01** | **8.06** | **18.39** | **19.18** | **34.93** | **14.87** | **19.09** |
| *Qwen2.5-Math-7B* | | | | | | | | | | |
| GRPO | 34.01 | 12.23 | 30.09 | 25.44 | **5.49** | 16.47 | 15.84 | 34.37 | 14.82 | 17.40 |
| CAPO | **30.50** | **6.42** | **18.94** | **18.62** | 5.82 | **16.03** | **13.15** | **23.17** | **12.38** | **14.11** |

*(1) Reduce the dependency on the size of the GenPRM.* We conducted additional experiments using smaller models, such as Qwen2.5-14B/32B-Instruct, as the GenPRM. These results demonstrate that CAPO does not rely on extremely large models (e.g., 72B), thereby significantly reducing inference latency. As shown in Tables 8, 6, and 5, CAPO consistently outperforms GRPO, RLOO, and the PRM-based method even when using these smaller, more efficient GenPRMs. This demonstrates that CAPO does not require prohibitive compute to be effective; the fine-grained signal provided by a moderately sized judge (e.g., 14B) is sufficient to surpass rule-based baselines.

*(2) Quantified Computational Overhead: CAPO vs. GRPO:* We provide and analyze the GPU hours of training Qwen2.5-Math-7B using a locally deployed Qwen2.5-14B-Instruct judge via vllm (the cost of deployment is also included in the compuatation of GPU hours below) and that using GRPO.

- **Rule-based Verification (GRPO)**: While simpler, this is not cost-free due to the overhead of external parsing libraries (e.g., math-verify) which lack thread safety. The average reward evaluation is **4.5s**, with a total per-step time of **47s**.

- **CAPO**: The average reward evaluation is **12.1s**, with a total per-step time of **55s**.

- **Total Cost**: For a full training cycle, GRPO consumed **272 GPU hours**, while CAPO consumed **331.2 GPU hours**. This represents a **21.7%** increase, which we believe is a manageable overhead given the performance gains and the reasons are as follows.

*(3) Efficiency of CAPO via Faster Convergence and Higher Upperbound* As detailed in Table 9, we showed the performance of GRPO and CAPO in different steps. We found that GRPO converge relatively slow and with limited gains on extended training. CAPO, however, achieve a high performance in a very beginning (step 60 and step 90) compared to GRPO which is also higher than the final training of GRPO and continue to improve. It is important to note that the 21.7% increase of GPU hours is based on the entire training but we actually can reduce the training steps for CAPO like using half steps (90 steps). In this situation, we still perform better than GRPO and using comparable or less GPU hours to GRPO.

We also show the further potential of CAPO to accelerate convergence. In our Qwen2.5-Math-1.5B experiments (Table 6), using a stronger judge (32B) allowed the model to reach optimal performance in just **1/3 of the training steps** required by the baseline. By enabling early stopping or reduce the total training steps, CAPO can, in practice, **reduce the total GPU hours** compared to rule-based methods to convergence, effectively offsetting the per-step overhead.

This approach is also well-aligned with an established and growing trend in the literature (Dong et al., 2023; Zhang et al., 2025a; Tang et al., 2025a; Yang et al., 2025): leveraging powerful, large-scale models—such as 70B-scale models like Qwen and Llama, or even closed-source APIs from Gemini, Grok, and OpenAI—to guide or refine policy models during training is a common and effective practice. For instance, many works employ larger models for data augmentation (Yan et al., 2025; Yang et al., 2025), trajectory refinement (Yan et al., 2025; Zhang et al., 2025a), or as sophisticated critics (Dong et al., 2023; Tang et al., 2025a). In this context, CAPO's single-pass

verification is a computationally efficient way to gain fine-grained process supervision. Furthermore, the practical implementation of CAPO is flexible. For researchers seeking to minimize local memory and computational costs, the GenPRM can be accessed via APIs, offloading the inference cost and making the framework highly accessible.

**Step Segmentation**    The current implementation of CAPO assigns credit at the step level, using \n \n as a delimiter following the widely accepted convention in the Process Reward Model (PRM) literature (Zhang et al., 2025b; Yang et al., 2024; Zheng et al., 2025). While most of the public open-source models like Qwen-series use this to seperate the steps, we here still discuss the granularity and the intrinsic advantage of our LLM-as-GenPRM approach: its inherent flexibility in handling task segmentation. Unlike rigid, rule-based systems that require predefined step structures, our framework can leverage the powerful multi-tasking and generalization capabilities of the LLMs (Brown et al., 2020; Ouyang et al., 2022; Chung et al., 2024). For our experiments, we wrapped the \n \n -delimited segments in <step_i> tags to construct the prompt. This was a deliberate choice for simplicity and reproducibility, especially since the Qwen models we used naturally structure their reasoning with this delimiter. However, this is not a fundamental requirement of CAPO. It is important to clarify that CAPO is a general framework for credit assignment, and the use of \n \n is merely a domain-specific implementation for mathematical reasoning. CAPO is agnostic to how steps are defined and can be easily adapted to other scenarios. In an Agentic workflow, the natural segmentation might be a "turn of interaction" or an "API call" rather than a newline character. CAPO can be directly applied to these segments without modification to the core algorithm.

The framework is powerful enough to support a more dynamic, end-to-end process. For instance, one could design a prompt that instructs the LLM-as-a-GenPRM to first autonomously segment the policy model's response into logical reasoning steps, and then, in the same inference pass, evaluate each of those self-identified steps. This would empower the GenPRM to adapt its segmentation strategy based on the content and structure of the specific response, moving beyond fixed delimiters to a more semantically meaningful division of logic. This potential for joint segmentation and evaluation showcases the true flexibility of using a general-purpose LLM as a verifier for adaptable credit assignment mechanism. It elegantly sidesteps the brittle nature of hard-coded parsing rules and opens the door to applying CAPO to a much wider range of tasks where reasoning steps may not be as cleanly formatted. The underlying framework readily accommodates future extensions towards more sophisticated, context-aware step identification, further demonstrating its power and versatility.

**Prompt Sensitivity of the GenPRM**    A potential concern for any method leveraging large language models is its sensitivity to the specific phrasing of the prompt. In CAPO, the GenPRM's performance could theoretically be influenced by the structure and wording of the verification prompt. While we found our chosen prompt (as detailed in Appendix F) to be effective and robust across different models and tasks, we acknowledge that prompt engineering can play a role in optimizing performance. A well-designed and well-consdiered prompt for GenPRM can probably improve the performance. We did not heavily tune the prompt used by LLM-as-a-GenPRM, but we indeed include some intuitive designs into the prompt, like asking the LLM to "verify each step of the solution independently, without relying on the correctness of the context before. This means that the current step cannot be considered as incorrect simply because it used the conclusion derived from the previous erroneous reasoning. ". We want the LLM to focus more on the quality or correctness of the currently verified step itself, rather than being influenced by the previous step. Preliminary experiments indicated that the model benefits from this design.

## D    IMPLEMENTATION DETAILS

As for the rule-based verifier, we use *Math-verify* to conduct an automatic judgment.

We follow Kazemnejad et al. for automatic step segmentation. Specifically, we partition the model-generated response into steps using \n \n as the delimiter. We then enclose each step with the markers <step_k> and </step_k> to denote the $k$-th step. Subsequently, we construct the prompt for LLM-as-GenPRM following the structure shown in the Prompt Section. The Case Study section provides a concrete example of a LLM-as-GenPRM generation.

All the hyperparameters during RL and SFT training could be found in Table 11. We follow the hyperparameters set in (Hugging Face, 2025) as a widely accepted setting.

Table 11: RL hyperparameter configurations for different models. Common settings are listed first, followed by the model-specific learning rates.

| Common Hyperparameter | Value |
|---|---|
| *SFT Stage* | |
| Learning Rate | $1 \times 10^{-5}$ |
| Batch Size | 80 |
| *RL Stage* | |
| PPO Mini-Epoch | 4 |
| Format Reward | 0.2 |
| KL Beta (`kl_beta`) | $1 \times 10^{-2}$ |
| Sampling Instances (`sampling_num`) | 16 |
| Global Batch Size | 192 |
| Rollout Micro Batch Size | 4 |
| LR Decay Style | Cosine |
| Rollout Top-p | 0.9 |
| Rollout Temperature | 0.7 |

| Learning Rates during RL Stage | | | | |
|---|---|---|---|---|
| Model | Qwen 1.5B | Qwen 7B | Llama 1B | Llama 3B |
| Learning Rate | $1 \times 10^{-7}$ | $2 \times 10^{-8}$ | $5 \times 10^{-8}$ | $2 \times 10^{-8}$ |

Table 12: Sampling Parameters for the LLM-as-a-GenPRM.

| Model | Temperature | Top-p | Max Tokens |
|---|---|---|---|
| Qwen2.5-72B-Instruct, Llama-3-70B-Instruct | 0.7 | 0.9 | 2048 |

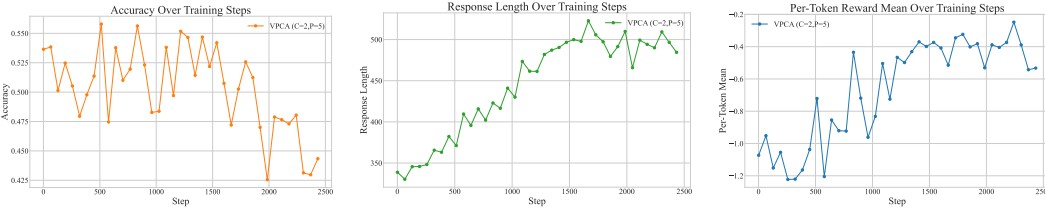

Figure 4: Training dynamics of CAPO (C=2,P=5) on Qwen2.5-1.5B. The figure plots the accuracy, response length, and mean per-token reward over training steps.

# E  ADDITIONAL EXPERIMENTS

**CAPO vs. Traditional PRMs**   A noteworthy finding from Llama-1B experiments in Table 8 is that PRM-based baselines performed relatively poorly, even worse than GRPO. We hypothesize this is due to distribution shift: open-source PRMs are typically trained on Qwen data, making them less effective judges for Llama. CAPO, which relies on a general-purpose instruction-following judge with GT, proved much more robust to this architecture shift, achieving superior performance.

**Detailed Discussion about Process-Oriented and Outcome-Oriented reward signals**   We explain and validate the underlying mechanism of interaction of these two reward signals below. A dominant outcome reward ensures rapid initial policy updates, enabling the model to acquire fundamental skills in the early stages. We posit that learning from the process is a more challenging objective. Early on, when the model's capability is limited, a correct process does not guarantee a correct outcome, indicating a weak correlation between the two. This could be validated in Figure 4.

Table 13: Comparison of results (Pass@1) across different methods and **different voting mechanisms** on Llama-3-1B-Instruct for mathematical (In-Distribution) reasoning tasks and general (Out-of-Distribution) reasoning tasks. $N=4$ critiques for CAPO with voting.

| Method | General Reasoning Tasks (OOD) | | | | Math Reasoning Tasks (ID) | | | | All Mean |
|---|---|---|---|---|---|---|---|---|---|
| | GPQA Diamond | arc_c | MMLU Pro | OOD Mean | MATH 500 | Olympiad Bench | AMC23 | Math Mean | |
| *Llama-3-1B-Instruct* | | | | | | | | | |
| SFT | 13.1 | 19.1 | **10.1** | 14.1 | 22.0 | 6.7 | 8.8 | 12.5 | 13.3 |
| GRPO-Rule | **28.8** | 21.6 | 4.5 | 18.3 | 27.1 | **7.0** | 6.2 | 13.4 | 15.9 |
| GRPO-GenORM | 25.6 | 22.0 | 4.4 | 17.4 | 27.2 | 6.3 | 7.6 | 13.7 | 15.6 |
| CAPO-Average | 22.9 | 29.3 | 5.3 | 19.2 | 26.8 | 5.4 | 10.4 | 14.2 | 16.7 |
| CAPO-Greedy | 25.6 | 28.5 | 5.7 | 19.9 | 26.3 | 5.7 | 9.8 | 13.9 | 16.9 |
| CAPO-Intersection | 25.3 | **34.0** | 6.2 | **21.8** | 28.7 | 6.5 | **12.0** | **15.7** | **18.8** |
| CAPO-Union | 21.2 | 29.6 | 5.5 | 18.8 | 28.1 | 5.5 | 8.5 | 14.0 | 16.4 |
| CAPO-MajorVote | 21.9 | 32.3 | 5.7 | 20.0 | 26.4 | 6.0 | 9.2 | 13.9 | 17.0 |

Table 14: Comparison of results (Pass@1) across different methods and **different voting mechanisms** on Qwen2.5-7B for mathematical (In-Distribution) reasoning tasks and general (Out-of-Distribution) reasoning tasks. $N=4$ critiques for CAPO with voting.

| Method | General Reasoning Tasks (OOD) | | | | Math Reasoning Tasks (ID) | | | | | All Mean |
|---|---|---|---|---|---|---|---|---|---|---|
| | GPQA Diamond | ARC_c | MMLU Pro | OOD Mean | MATH 500 | Olympiad Bench | AIME24 | AMC23 | Math Mean | |
| *Qwen-2.5-7B* | | | | | | | | | | |
| SFT | **23.7** | 67.7 | 32.5 | 41.3 | 60.7 | 28.1 | 3.3 | 33.0 | 31.3 | 34.7 |
| GRPO-Rule | 22.4 | 68.1 | 32.7 | 41.0 | 59.9 | 27.2 | 3.6 | 34.1 | 31.2 | 34.5 |
| GRPO-GenORM | 23.6 | 68.0 | **32.9** | **41.5** | 59.2 | 27.6 | 4.0 | 33.4 | 31.1 | 34.5 |
| CAPO-Average | 23.2 | 66.9 | 32.8 | 41.0 | 62.6 | **32.0** | 8.3 | 33.9 | 34.2 | 36.7 |
| CAPO-Greedy | 21.5 | 68.2 | 32.9 | 40.9 | 62.4 | 31.3 | 9.7 | 34.2 | 34.4 | 36.8 |
| CAPO-Intersection | 22.2 | **68.3** | 32.9 | 41.2 | 62.5 | 30.5 | 7.1 | 34.6 | 33.7 | 36.3 |
| CAPO-Union | 21.4 | 67.2 | 32.7 | 40.4 | 62.4 | 31.3 | 9.1 | **37.1** | **35.0** | 36.8 |
| CAPO-MajorVote | 22.7 | 67.0 | 32.8 | 40.8 | **63.4** | 31.0 | **10.8** | 34.1 | 34.8 | **37.0** |

We could see that in this reward configuration focusing on process, the model learn to be verbose instead of learning to improve the outcome. Therefore, overemphasizing the process while the model is still weak is not meaningful; accumulating correct process steps might not lead to a correct result and can even slow down the overall optimization which is stated in our first Finding in the Method Section. However, the process reward becomes crucial in the later stages of optimization. Even a small process reward can be beneficial as shown in Table 7, introducing a small fraction of process signal(C=2,P=0.1) can lead to the second best performance. It helps to differentiate and prioritize samples with better reasoning among a large pool of responses that all achieve the same outcome. This improves sample efficiency and promotes more effective learning. When optimization based on the outcome reward hits a plateau (i.e., with many samples receiving identical outcome rewards), the process-oriented signal becomes critically important. It helps to differentiate and prioritize samples with better reasoning among numerous responses that achieve the same outcome and make the model's continued training more effective, which leads to our second Finding in the previous section.

**Discussion about Different Voting Mechanisms**    We have introduced the two most representative voting mechanisms in the previous section. And here, we supplement the introduction of the rest three voting mechanisms involved in our experiments below.

- **Greedy:** This is the simplest strategy, where we generate only a single critique ($k = 1$) using greedy decoding. The final set of erroneous indices is simply $\mathcal{S}_i^* = \mathcal{S}_{i,1}$. It is sensitive to noise and misjudgment.

- **Union Vote:** This approach maximizes recall by flagging any step identified as erroneous by at least one critique: $\mathcal{S}_i^{\cup} = \bigcup_{j=1}^{k} \mathcal{S}_{i,j}$. It is comprehensive in capturing potential errors

Table 15: Performance comparison of Qwen-2.5-7B model variants based on the CAPO-Intersection method. Results are presented in percentages (%). **'w/o gt'** denotes the model trained without ground truth, while **'w gt'** indicates the model trained with ground truth.

| Method | General Reasoning Tasks (OOD) | | | | Math Reasoning Tasks (ID) | | | | | All Mean |
|---|---|---|---|---|---|---|---|---|---|---|
| | GPQA Diamond | ARC_c | MMLU Pro | OOD Mean | MATH 500 | Olympiad Bench | AIME24 | AMC23 | Math Mean | |
| CAPO-w/o gt | **24.4** | **69.9** | **36.0** | **43.5** | 62.1 | 30.6 | 4.0 | 32.8 | 32.4 | 37.9 |
| CAPO-w gt | 24.2 | 69.1 | 35.4 | 42.9 | **62.3** | **31.2** | **4.2** | **34.5** | **33.1** | **38.0** |

Table 16: Performance comparison of different GRPO-GenORM configurations on OOD benchmarks. Scores are rounded to three decimal places. The highest score in each column is **bolded**.

| Model | GPQA Diamond | ARC_c | MMLU Pro | OOD Mean |
|---|---|---|---|---|
| GRPO-GenORM ($N$=2) | 15.3 | 60.6 | 20.4 | 32.1 |
| GRPO-GenORM ($N$=4) | 16.7 | **61.5** | 20.8 | 33.0 |
| GRPO-GenORM ($N$=8) | **18.7** | 61.2 | **20.9** | **33.6** |

but is more susceptible to false positives, as a single faulty critique can introduce an error into the final set.

- **Average:** Instead of a binary decision, this strategy assigns a continuous penalty weight to each step based on frequency. The penalty weight for a step index $s$ is calculated as $w(s) = \text{count}(s)/k$. This allows for a more nuanced credit assignment where consistently flagged errors receive a stronger penalty signal than those identified less frequently, avoiding hard thresholds.

A key finding from our experiments, detailed in Table 13 and Table 14, is that all variants of CAPO consistently outperform the GRPO baseline in mean scores. This result strongly validates our core hypothesis: the integration of reliable, fine-grained credit assignment is the primary driver of performance gains, enhancing both mathematical and general reasoning capabilities regardless of the specific aggregation strategy. Beyond this, a deeper analysis of the different voting mechanisms reveals a nuanced and insightful relationship between the strictness of the reward signal and the capability of the backbone model.

We observe a distinct pattern: **the optimal voting strategy appears to correlate with model scale.** For the smaller Qwen2.5-1.5B and Llama-3-1B-Instruct models, the Intersection voting mechanism yields the best overall performance. This mechanism, which requires consistent agreement from all critiques of GenPRM to assign a positive reward, provides the highest-precision learning signal. We posit that for smaller, less capable models, this strict criterion acts as a powerful corrective pressure. It forces the model to learn only the most robust and unequivocally correct reasoning steps, effectively filtering out noisy or partially correct pathways that could otherwise be spuriously reinforced. This high-precision guidance is particularly effective at building a solid reasoning foundation and enhancing generalization, as evidenced by its top performance on OOD benchmarks for these models.

Conversely, on the larger and more capable Qwen2.5-7B backbone, the mechanisms with looser constraints—specifically majority-based voting and union-based voting—emerge as the top performers. We hypothesize that for stronger models tackling more complex problems, an overly strict rule like Intersection can be suboptimal. A single critique from GenPRM might incorrectly flag a valid or complex reasoning step as faulty (a false negative), which would prevent exploration. Majority-based or union-based voting strikes a more effective balance between precision and recall. These mechanisms tolerate minor disagreements among the different critiques, leveraging the "wisdom of the crowd" to reward trajectories that are directionally correct, even if they contain unexpected steps that not all verifiers approve. This allows the model to explore a more diverse set of valid reasoning strategies, which is critical for solving higher-difficulty problems.

In summary, the choice of voting mechanism represents a tunable trade-off between signal precision and exploratory freedom. While stricter mechanisms are beneficial for building foundational correctness in smaller models, more lenient, consensus-based approaches better facilitate the learning of complex reasoning in larger models. This adaptability underscores the flexibility and robustness of the CAPO framework.

**Will the incorporation of Ground Truth Solution impact the performance of CAPO?** In this section, we investigate the impact of including ground-truth (GT) solutions on the discriminative performance of the GenPRM. The empirical results, summarized in Table 15, demonstrate that the inclusion of ground-truth solutions does not always guarantee a notable advantage to our method. On the contrary, we observe a slight degradation in general reasoning performance, suggesting that GT solutions may not be inherently beneficial in this context. This phenomenon is acceptable, especially when considering the potential for the gap of the model distribution between the GT data and the policy model's solution space. This is particularly true when a significant distributional discrepancy exists between the GT solutions and the outputs generated by the policy model. Consequently, instead of providing a clear learning signal, these GT solutions may introduce noise that confuses the GenPRM and ultimately hinders its ability to make accurate and reliable discriminative judgments.

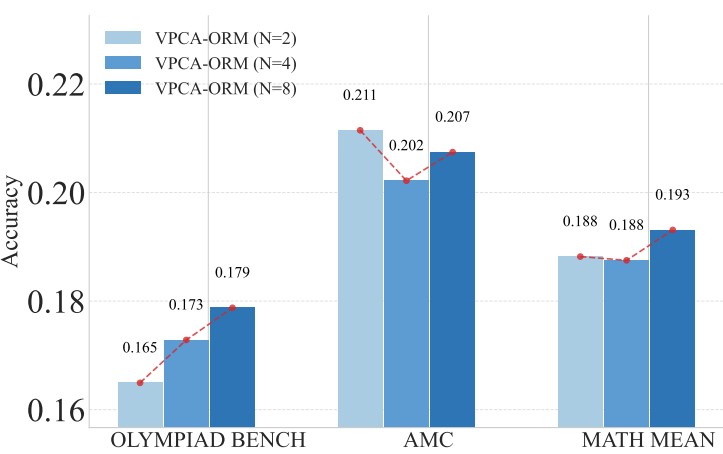

Figure 5: Effect of the number of samples generated by GenORM on GRPO Performance. The plot illustrates the performance of GRPO when using a varying number of samples, $N \in \{2, 4, 8\}$ from the GenORM. Increasing the sample size $N$ leads to unstable performance increase and the performance increase is relatively small compared to using LLM-as-GenRRM in CAPO.

**Statistical Validation and Variance Analysis** To provide a more comprehensive analysis of the results presented in Table 3 and Table 2, we now examine the variance and statistical stability of our findings in Table 18 and Table 17. Presenting standard deviations or confidence intervals directly within the main results table would compromise its readability due to the density of information. Therefore, we present a detailed breakdown in a separate table for clarity. As can be seen in the tables, our proposed method not only achieves superior mean performance but also exhibits a relatively low standard deviation across most tasks compared to the baselines. The small variance underscores the robustness of our method, confirming that the observed improvements are statistically significant and not due to random chance.

**Discussion about GenORM and GenPRM** As discussed in Yang et al. (2024), we combine the rule-based verifier and generative outcome-based verifier using the reward shaping technique (Ng et al., 1999; Wiewiora et al., 2003) as below:

$$R(r_v, r_m) = \sigma(\alpha \cdot r_m) + (\beta \cdot r_v - 1), \tag{3}$$

where $R(r_v, r_m)$ be the final reward, and $r_v, r_m \in \{1, 0\}$. Here, $r_v = 1$ indicates a correct response according to the rule-based verifier, and $r_m = 1$ indicates a high-quality response according to the GenPRM.

Table 17: Comparison of results (Pass@1) (**with variance**) across different methods on Llama-3-1B and Qwen2.5-1.5B for mathematical (In-Distribution) and general (Out-of-Distribution) reasoning tasks. N=4 critiques for CAPO with voting.

| Method | General Reasoning Tasks (OOD) | | | | Math Reasoning Tasks (ID) | | | | All Mean |
|---|---|---|---|---|---|---|---|---|---|
| | GPQA Diamond | arc_c | MMLU Pro | OOD Mean | MATH 500 | Olympiad Bench | AMC23 | Math Mean | |
| *Llama-3-1B* | | | | | | | | | |
| SFT | 13.1 ± 1.8 | 19.1 ± 0.6 | **10.1** ± 0.2 | 14.1 ± 0.8 | 22.0 ± 1.7 | 6.7 ± 0.0 | 8.8 ± 3.6 | 12.5 ± 1.1 | 13.3 ± 0.3 |
| GRPO-Rule | **28.8** ± 0.0 | 21.6 ± 0.1 | 4.5 ± 0.1 | 18.3 ± 0.1 | 27.1 ± 0.4 | **7.0** ± 0.9 | 6.2 ± 0.8 | 13.4 ± 0.6 | 15.9 ± 0.4 |
| GRPO-GenORM | 25.6 ± 0.3 | 22.0 ± 0.3 | 4.4 ± 0.0 | 17.4 ± 0.1 | 27.2 ± 0.7 | 6.3 ± 0.3 | 7.6 ± 0.9 | 13.7 ± 0.2 | 15.5 ± 0.2 |
| CAPO-Greedy | 25.6 ± 2.1 | 28.5 ± 0.1 | 5.7 ± 0.1 | 19.9 ± 0.7 | 26.3 ± 0.8 | 5.7 ± 0.2 | 9.8 ± 0.6 | 13.9 ± 0.4 | 16.9 ± 0.3 |
| CAPO-Intersection | 25.3 ± 0.0 | **34.0** ± 0.5 | 6.2 ± 0.1 | **21.8** ± 0.2 | **28.7** ± 0.8 | 6.5 ± 0.4 | **12.0** ± 0.4 | **15.7** ± 0.1 (+2.0%) | **18.8** ± 0.1 (+2.9%) |
| CAPO-MajorVote | 21.9 ± 0.6 | 32.3 ± 0.2 | 5.7 ± 0.0 | 20.0 ± 0.2 | 26.4 ± 0.6 | 6.0 ± 0.6 | 9.2 ± 1.5 | 13.9 ± 0.5 | 16.9 ± 0.3 |
| *Qwen-2.5-1.5B* | | | | | | | | | |
| SFT | 16.0 ± 0.3 | **61.1** ± 0.5 | 20.9 ± 0.1 | 32.7 ± 0.3 | 47.5 ± 1.0 | 16.0 ± 0.2 | **26.1** ± 1.7 | 29.9 ± 0.4 | 31.3 ± 0.1 |
| GRPO-Rule | 20.4 ± 1.3 | 60.6 ± 0.6 | 20.7 ± 0.2 | 33.9 ± 0.6 | 48.9 ± 0.4 | 17.0 ± 0.7 | | 28.8 ± 0.6 | 31.3 ± 0.4 |
| GRPO-GenORM | 18.2 ± 0.9 | 60.8 ± 0.3 | 20.9 ± 0.1 | 33.3 ± 0.4 | 49.2 ± 0.4 | 17.4 ± 0.7 | 20.2 ± 2.0 | 28.9 ± 0.3 | 31.1 ± 0.1 |
| CAPO-Greedy | 14.8 ± 0.6 | 59.8 ± 0.4 | **21.9** ± 0.2 | 32.2 ± 0.0 | 49.3 ± 0.6 | 18.1 ± 0.8 | 22.4 ± 0.3 | 29.9 ± 0.3 | 31.1 ± 0.2 |
| CAPO-Intersection | **20.5** ± 0.3 | 60.0 ± 0.4 | 21.8 ± 0.1 | **34.1** ± 0.5 | 48.7 ± 0.1 | **18.8** ± 0.7 | 24.0 ± 0.8 | **30.5** ± 0.4 (+0.6%) | **32.3** ± 0.1 (+1.0%) |
| CAPO-MajorVote | 19.9 ± 1.5 | 60.4 ± 0.8 | 21.6 ± 0.1 | 34.0 ± 0.6 | **49.7** ± 0.4 | 18.6 ± 0.2 | 22.0 ± 0.2 | 30.1 ± 0.0 | 32.0 ± 0.3 |

Table 18: Comparison of results (Pass@1) (**with variance**) across different methods on Llama-3-3B and Qwen2.5-7B for mathematical (In-Distribution) and general (Out-of-Distribution) reasoning tasks. N=4 critiques for CAPO with voting.

| Method | General Reasoning Tasks (OOD) | | | | Math Reasoning Tasks (ID) | | | | | All Mean |
|---|---|---|---|---|---|---|---|---|---|---|
| | GPQA Diamond | ARC_c | MMLU Pro | OOD Mean | MATH 500 | Olympiad Bench | AIME24 | AMC23 | Math Mean | |
| *Llama-3-3B* | | | | | | | | | | |
| SFT | 19.7 ± 1.0 | 56.1 ± 0.2 | 21.0 ± 0.1 | 32.3 ± 0.7 | 43.2 ± 0.8 | 12.7 ± 0.4 | 6.7 ± 2.1 | 10.8 ± 1.1 ± 0.6 | 18.4 ± 0.4 | 24.3 ± 0.6 |
| GRPO-Rule | 19.9 ± 0.7 | 61.2 ± 0.2 | **32.4** ± 0.0 | 37.8 ± 0.1 | 44.6 ± 0.0 | 15.0 ± 0.0 | **10.0** | 16.9 ± 0.0 | 21.6 ± 0.0 | 28.6 ± 0.2 |
| GRPO-GenORM | 22.2 ± 1.0 | 62.2 ± 1.1 | 31.8 ± 00.1 | 38.7 ± 0.6 | **46.6** ± 0.5 | 14.8 ± 0.4 | 3.3 ± 0.3 | 20.0 ± 1.2 | 21.2 ± 0.3 | 28.5 ± 0.4 |
| CAPO-Greedy | **24.2** ± 0.5 | 61.9 ± 0.3 | 31.7 ± 0.1 | **39.3** ± 0.1 | 45.6 ± 0.4 | 14.1 ± 0.5 | 3.3 | 19.3 ± 0.3 | 20.6 ± 0.1 | 28.6 ± 0.2 |
| CAPO-Intersection | 23.2 ± 1.0 | **63.2** ± 1.2 | 31.5 ± 0.1 | **39.3** ± 0.6 | 44.3 ± 0.6 | 15.6 ± 0.5 | 7.8 | 20.3 ± 1.2 | 22.0 ± 0.4 | 29.4 ± 0.4 |
| CAPO-MajorVote | 20.9 ± 1.0 | 62.4 ± 0.1 | 31.5 ± 0.0 | 38.3 ± 0.7 | 44.6 ± 0.0 | **16.7** ± 0.1 | 6.7 | **26.8** ± 0.3 | **23.7** ± 0.1 (+2.1%) | **30.4** ± 0.2 (+1.8%) |
| *Qwen-2.5-7B* | | | | | | | | | | |
| SFT | **23.7** ± 2.6 | 67.7 ± 0.2 | 32.5 ± 0.0 | **41.3** ± 0.8 | 60.7 ± 0.8 | 28.1 ± 0.8 | 3.3 ± 1.1 | 33.0 ± 0.2 | 31.3 ± 0.1 | 34.7 ± 0.4 |
| GRPO-Rule | 22.4 ± 0.3 | 68.1 ± 0.2 | 32.7 ± 0.0 | 41.0 ± 0.1 | 59.9 ± 0.9 | 27.2 ± 0.1 | 3.6 ± 0.6 | 34.1 ± 0.6 | 31.2 ± 0.1 | 34.5 ± 0.1 |
| GRPO-GenORM | 23.6 ± 0.6 | 68.0 ± 0.2 | **32.9** ± 0.2 | **41.5** ± 0.3 | 59.2 ± 0.4 | 27.6 ± 0.5 | 4.0 ± 0.7 | 33.4 ± 0.5 | 31.1 ± 0.1 | 34.5 ± 0.1 |
| CAPO-Greedy | 21.5 ± 0.6 | 68.2 ± 0.3 | 32.9 ± 0.1 | 40.9 ± 0.3 | 62.4 ± 0.9 | **31.3** ± 1.0 | 9.7 ± 0.6 | 34.2 ± 0.3 | 34.4 ± 0.4 | **36.8** ± 0.3 |
| CAPO-Intersection | 22.2 ± 1.3 | **68.3** ± 0.6 | 32.9 ± 0.1 | 41.2 ± 0.7 | 62.5 ± 0.8 | 30.5 ± 0.1 | 7.1 ± 0.5 | **34.6** ± 0.9 | 33.7 ± 0.5 | 36.3 ± 0.3 |
| CAPO-MajorVote | 22.7 ± 1.0 | 67.0 ± 0.4 | 32.8 ± 0.1 | 40.8 ± 0.3 | **63.4** ± 0.0 | 31.0 ± 0.0 | **10.8** ± 0.5 | 34.1 ± 0.0 | **34.8** ± 0.0 (+3.5%) | **37.0** ± 0.2 (+2.3%) |

As shown in Figure 5 and Table 16, the GRPO using GenRM as outcome verifier exhibits a much less stable and beneficial response to scaling $N$. While there is a slight overall improvement, the trend is less evident. For instance, on Math benchmarks in Figure 5, performance on AMC is unstable, and the gains on the Math Mean are marginal compared to CAPO. This suggests that simply increasing the number of samples for an outcome-based method offers limited benefits, as it can't offer fine-grained credit assignment for model update.

**Case Study of Different Methods** The response generated by the model trained by CAPO demonstrates a significant improvement in robustness, clarity, and efficiency over that of GRPO. The response generated by GRPO attempts to solve the problem by converting the line equations into a standard symmetric form, a process that unnecessarily introduces complex fractions. This strategy not only increases the cognitive load but also elevates the risk of both conceptual and arithmetic errors, ultimately leading to an incorrect and convoluted final expression.

In contrast, CAPO adopts a more fundamental and elegant strategy by determining the simplest integer-based direction vector for each line. By treating the equations as a system of ratios and identifying a vector that satisfies them with integer components, CAPO circumvents fractional arithmetic entirely. This simplification cascades through all subsequent steps, particularly the dot product calculation, which becomes trivial and far less error-prone.

Consequently, our method's superiority lies not merely in achieving the correct answer, but in its inherent efficiency and reduced susceptibility to common errors, reflecting a more direct and intuitive application of vector principles.

**Case Study of Internal Variations in Rollout Responses** To concretely illustrate *the inadequacy of binary, outcome-based rewards*, we present a case study comparing four distinct model-generated responses during RL training to the same mathematical problem. These examples highlight how trajectories with identical final outcomes can possess vastly different internal reasoning qualities, a nuance that coarse-grained reward signals fail to capture.

We first examine two responses that arrive at the correct answer of 20. The correct response in Figure F demonstrates a flawless, direct, and correct reasoning process. In contrast, the response in Figure F reaches the same correct answer through a spurious and convoluted line of reasoning, as highlighted in red. Under a standard RLVR framework, both responses would receive an identical positive reward, rendering the model unable to distinguish between sound logic and fortunate coincidence.

Next, we compare two responses that fail to produce the correct answer. The response in Figure F demonstrates a strong understanding of the core logic (i.e., that the numbers must be equal), and its failure is isolated to a minor off-by-one calculation error at the end. Conversely, the response in Figure F is fundamentally flawed, employing a completely incorrect and irrelevant problem-solving strategy from the outset. A binary reward scheme would assign the same penalty to both, failing to credit the largely correct reasoning in the third case while not sufficiently penalizing the complete conceptual failure in the fourth.

This case study underscores that a single binary reward signal obscures critical, process-level information. This lack of differentiated feedback prevents the model from learning to favor robust reasoning pathways over flawed or lucky ones, motivating the need for a more granular credit assignment mechanism.

## F    PROMPT

**Prompt for Generative Reward model.** Below is the prompt for the generative reward model.

---

**Prompt for Generative Reward Model**

You are a math teacher.  Use [Ground Truth Solution] to find any erroneous step in [Solution To Judge].  And the solution to be checked is separated with steps like "<step_i>...</step_i>" where i is the index of the step.  First, determine whether the solution is correct based on [Ground Truth Solution] and write it in the form "Judgment: Is the solution correct (Yes/No)?  X", where X is either Yes or No.  If the solution is incorrect, review, analyze, and verify each step of the solution independently, without relying on the correctness of the context before.  This means that the current step cannot be considered as incorrect simply because it used the conclusion derived from the previous erroneous reasoning.  Then provide the list of the indices of error steps separated by a comma.  And the format is as follows:  <incorrect_steps>...(the index of the incorrect steps)... </incorrect_steps>. Please give your answer directly.

---

**Training Prompt.** We use the following prompt to conduct SFT and RL training.

---

**Training Prompt**

**System**
Please reason step by step, and put your final answer within \boxed{}.
**User**
{Question Content}

---

**Question**
The set of points $(x, y, z)$ that satisfy

$$2x = 3y = -z$$

is a line.
The set of points $(x, y, z)$ that satisfy

$$6x = -y = -4z$$

is another line. Find the angle between these lines, in degrees.

---

**Case Study of CAPO**

For the first line, we can express the equations as:

$$2x = 3y = -z$$

This implies:

$$x = \frac{3}{2}y, \quad z = -2x = -3y$$

Thus, the direction vector of the first line is $\begin{pmatrix} 3 \\ 2 \\ -3 \end{pmatrix}$.

For the second line, we can express the equations as:

$$6x = -y = -4z$$

This implies:

$$y = -6x, \quad z = -\frac{3}{2}x$$

Thus, the direction vector of the second line is $\begin{pmatrix} 1 \\ -6 \\ -3 \end{pmatrix}$.

To find the angle $\theta$ between these lines, we use the dot product formula:

$$\cos \theta = \frac{\begin{pmatrix} 3 \\ 2 \\ -3 \end{pmatrix} \cdot \begin{pmatrix} 1 \\ -6 \\ -3 \end{pmatrix}}{\left\| \begin{pmatrix} 3 \\ 2 \\ -3 \end{pmatrix} \right\| \left\| \begin{pmatrix} 1 \\ -6 \\ -3 \end{pmatrix} \right\|}$$

Calculating the dot product:

$$3 \cdot 1 + 2 \cdot (-6) + (-3) \cdot (-3) = 3 - 12 + 9 = 0$$

Calculating the magnitudes:

$$\left\| \begin{pmatrix} 3 \\ 2 \\ -3 \end{pmatrix} \right\| = \sqrt{3^2 + 2^2 + (-3)^2} = \sqrt{14}$$

$$\left\| \begin{pmatrix} 1 \\ -6 \\ -3 \end{pmatrix} \right\| = \sqrt{1^2 + (-6)^2 + (-3)^2} = \sqrt{46}$$

Thus,

$$\cos \theta = \frac{0}{\sqrt{14} \cdot \sqrt{46}} = 0$$

Therefore, $\theta = \boxed{90°}$.

**Case Study of GRPO (Failed Response)**

For the first line, we can express the equation as:

$$\frac{x}{1} = \frac{y}{3/2} = \frac{z}{-1/2}.$$

Thus, a direction vector for this line is $\begin{pmatrix} 1 \\ 3/2 \\ -1/2 \end{pmatrix}$.

For the second line, we can express the equation as:

$$\frac{x}{-1/6} = \frac{y}{1} = \frac{z}{1/4}.$$

Thus, a direction vector for this line is $\begin{pmatrix} -1/6 \\ 1 \\ 1/4 \end{pmatrix}$.

To find the angle $\theta$ between these lines, we use the dot product formula:

$$\cos\theta = \frac{\begin{pmatrix} 1 \\ 3/2 \\ -1/2 \end{pmatrix} \cdot \begin{pmatrix} -1/6 \\ 1 \\ 1/4 \end{pmatrix}}{\left\| \begin{pmatrix} 1 \\ 3/2 \\ -1/2 \end{pmatrix} \right\| \left\| \begin{pmatrix} -1/6 \\ 1 \\ 1/4 \end{pmatrix} \right\|}.$$

Calculating the dot product:

$$\begin{pmatrix} 1 \\ 3/2 \\ -1/2 \end{pmatrix} \cdot \begin{pmatrix} -1/6 \\ 1 \\ 1/4 \end{pmatrix} = 1 \cdot (-1/6) + (3/2) \cdot 1 + (-1/2) \cdot (1/4) = -1/6 + 3/2 - 1/8 = 11/8.$$

Calculating the magnitudes:

$$\left\| \begin{pmatrix} 1 \\ 3/2 \\ -1/2 \end{pmatrix} \right\| = \sqrt{1^2 + (3/2)^2 + (-1/2)^2} = \sqrt{1 + 9/4 + 1/4} = \sqrt{11/2},$$

$$\left\| \begin{pmatrix} -1/6 \\ 1 \\ 1/4 \end{pmatrix} \right\| = \sqrt{(-1/6)^2 + 1^2 + (1/4)^2} = \sqrt{1/36 + 1 + 1/16} = \sqrt{17/12}.$$

Thus,

$$\cos\theta = \frac{11/8}{\sqrt{11/2} \cdot \sqrt{17/12}} = \frac{11/8}{\sqrt{187/24}} = \frac{11/8}{\sqrt{187}/\sqrt{24}} = \frac{11 \cdot \sqrt{24}}{8 \cdot \sqrt{187}}.$$

Solving for $\theta$:

$$\theta = \cos^{-1}\left(\frac{11 \cdot \sqrt{24}}{8 \cdot \sqrt{187}}\right).$$

Conclusion: $\boxed{\theta = \cos^{-1}\left(\frac{11 \cdot \sqrt{24}}{8 \cdot \sqrt{187}}\right)}$.

**The response with correct answer and correct reasoning process.**

**User**

Mary and Jenna are playing a game in which each girl rolls a twenty-sided die with numbers from 1 to 20 on the faces. If the number on the die that Mary rolls is a factor of the number on the die that Jenna rolls, Mary wins. If the number on the die that Jenna rolls is a factor of the number on the die that Mary rolls, Jenna wins. For how many possible rolls would both girls win?

**Assistant**

To solve this problem, we need to find the number of pairs $(m, j)$ where $m$ is the number rolled by Mary and $j$ is the number rolled by Jenna, such that $m$ is a factor of $j$ and

$j$ is a factor of $m$. This implies that $m$ and $j$ must be equal, since $m$ being a factor of $j$ and $j$ being a factor of $m$ can only happen if $m = j$.

Given that both Mary and Jenna are rolling a twenty-sided die, the possible values for $m$ and $j$ range from 1 to 20. Therefore, there are 20 possible outcomes where both $m$ and $j$ are equal.

Hence, the number of possible rolls where both girls win is 20 .

---

**The response with correct answer and the incorrect reasoning process.**

**User**

Mary and Jenna are playing a game in which each girl rolls a twenty-sided die with numbers from 1 to 20 on the faces. If the number on the die that Mary rolls is a factor of the number on the die that Jenna rolls, Mary wins. If the number on the die that Jenna rolls is a factor of the number on the die that Mary rolls, Jenna wins. For how many possible rolls would both girls win?

**Assistant**

To solve this problem, we first need to understand the conditions under which both Mary and Jenna can win. For Mary to win, the number on her die must be a factor of the number on Jenna's die. Conversely, for Jenna to win, the number on her die must be a factor of the number on Mary's die. The only way both can win simultaneously is if the number on both dice is the same, because a number is always a factor of itself.

Given that each die has numbers from 1 to 20, we need to find how many numbers in this range have at least one other number in the same range that is a factor of them. This means we are looking for numbers that have at least one other number in the range that divides them without a remainder.
- The number 1 is a factor of every number, so it cannot be the number that both girls roll to win because there is no other number in the range that is a factor of 1.
- The numbers 2, 3, 4, 5, 6, 7, 8, 9, 10, 11, 12, 13, 14, 15, 16, 17, 18, 19, and 20 each have at least one other number in the range that is a factor of them (for example, 2 is a factor of 4, 6, 8, 10, 12, 14, 16, 18, and 20; 3 is a factor of 6, 9, 12, 15, and 18; and so on).

Therefore, there are 20 numbers in the range from 1 to 20 that have at least one other number in the same range that is a factor of them. Since both girls can roll any of these numbers and win, there are 20 possible rolls where both girls would win.

---

**The response with wrong answer and few incorrect reasoning process.**

**User**

Mary and Jenna are playing a game in which each girl rolls a twenty-sided die with numbers from 1 to 20 on the faces. If the number on the die that Mary rolls is a factor of the number on the die that Jenna rolls, Mary wins. If the number on the die that Jenna rolls is a factor of the number on the die that Mary rolls, Jenna wins. For how many possible rolls would both girls win?

**Assistant**

To answer this question, we need to find the circumstances under which Mary and Jenna both win. Let's denote Mary's roll as M and Jenna's roll as J.

The winning conditions are as follows: Mary wins if M is a factor of J (M — J). Jenna wins if J is a factor of M (J — M).

For both girls to win, these two conditions must be met simultaneously. For positive integers, if M is a factor of J and J is also a factor of M, the only possibility is that M = J.

Therefore, our task is to count how many pairs (M, J) satisfy the condition M = J. That is, both girls roll the exact same number.

The faces of the die are numbered from 1 to 20. So, we need to calculate how many possible numbers are in this range.

- The numbers on the die are from 1 to 20. This is a sequence of consecutive integers.
- To count the number of integers in this range, we can subtract the minimum value from the maximum value.
- Therefore, the number of possible outcomes is: $20 - 1 = 19$.

Since they can both win only when they roll the same number, and there are a total of 19 such possible numbers, there are 19 satisfying combinations. These combinations are (2,2), (3,3), (4,4), ..., (20,20).

Therefore, the total number of possible rolls where both girls win is $\boxed{19}$.

---

**The response with wrong answer and totally wrong reasoning process.**

**User**

Mary and Jenna are playing a game in which each girl rolls a twenty-sided die with numbers from 1 to 20 on the faces. If the number on the die that Mary rolls is a factor of the number on the die that Jenna rolls, Mary wins. If the number on the die that Jenna rolls is a factor of the number on the die that Mary rolls, Jenna wins. For how many possible rolls would both girls win?

**Assistant**

To solve this problem, we need to count the number of ways in which both Mary and Jenna can win under the given conditions.

1. **Case 1: Jenna rolls a prime number.** - A prime number has exactly two factors: 1 and itself. - Since there are 8 prime numbers between 1 and 20 (2, 3, 5, 7, 11, 13, 17, 19), and Mary can roll any number that is a factor of Jenna's roll, Mary can only win if she rolls a 1. - Therefore, there are 8 ways in this case where Mary wins.

2. **Case 2: Jenna rolls a composite number that is not a perfect square.** - Composite numbers between 1 and 20 that are not perfect squares are 4, 6, 8, 10, 12, 14, 15, 18, and 20. - For each of these numbers, there are two factors (other than 1 and the number itself) that Mary can roll to win. For example, if Jenna rolls a 4, Mary can win by rolling a 2; if Jenna rolls a 6, Mary can win by rolling a 2 or a 3, and so on. - There are 9 such numbers, and for each, Mary has 2 choices, resulting in $9 \times 2 = 18$ ways.

3. **Case 3: Jenna rolls a perfect square.** - Perfect squares between 1 and 20 are 1, 4, 9, and 16. - If Jenna rolls a 1, Mary can win by rolling any number (20 choices), but Jenna also wins if Mary rolls a 1, resulting in $20 - 1 = 19$ ways. - If Jenna rolls a 4, Mary can win by rolling a 1 or a 2 (2 choices), but Jenna wins if Mary rolls a 4, resulting in $2 - 1 = 1$ way. - If Jenna rolls a 9, Mary can win by rolling a 1 or a 3 (2 choices), but Jenna wins if Mary rolls a 9, resulting in $2 - 1 = 1$ way. - If Jenna rolls a 16, Mary can win by rolling a 1, 2, 4, or 8 (4 choices), but Jenna wins if Mary rolls a 16, resulting in $4 - 1 = 3$

ways. - Adding these up, we get $19 + 1 + 1 + 3 = 24$ ways.

4. **Total number of winning combinations:** - Adding the numbers from all cases, we get $8 + 18 + 24 = 50$ ways.

Therefore, the total number of possible rolls where both girls win is $\boxed{50}$.

Case Study of LLM-as-GenPRM when Response is Correct

You are a math teacher. Use [Ground Truth Solution] to find any erroneous step in [Solution To Judge]. And the solution to be checked is separated with steps like "<step_i>...</step_i>" where i is the index of the step. First, determine whether the solution is correct based on [Ground Truth Solution] and write it in the form "Judgment: Is the solution correct (Yes/No)? X", where X is either Yes or No. If the solution is incorrect, review, analyze, and verify each step of the solution independently, without relying on the correctness of the context before. This means that the current step cannot be considered as incorrect simply because it used the conclusion derived from the previous erroneous reasoning. Then provide the list of the indices of error steps separated by a comma. And the format is as follows:

<incorrect_steps>
...(the index of the incorrect steps)...
</incorrect_steps>.

[Question]
If $a$ is a constant such that $4x^2 - 12x + a$ is the square of a binomial, then what is $a$? Please reason step by step, and put your final answer within boxed.

[Ground Truth Solution] If $4x^2 - 12x + a$ is the square of a binomial, then the binomial has the form $2x + b$ for some number $b$, since $(2x)^2 = 4x^2$. So, we compare $(2x + b)^2$ to $4x^2 - 12x + a$. Expanding $(2x + b)^2$ gives

$$(2x + b)^2 = 4x^2 + 4bx + b^2.$$

Equating the linear term of this to the linear term of $4x^2 - 12x + a$, we have $4bx = -12x$, so $b = -3$. Thus, $a = b^2 = \boxed{9}$.

[Solution To Judge]
<step_0>
## Step 1: Recognize the form of a perfect square trinomial A perfect square trinomial can be written in the form $(px + q)^2$, which expands to $p^2x^2 + 2pqx + q^2$. Here, we need to match this form with $4x^2 - 12x + a$.
</step_0>
<step_1>
## Step 2: Identify the coefficients of the perfect square trinomial The coefficient of $x^2$ in the perfect square trinomial is $p^2$, the coefficient of $x$ is $2pq$, and the constant term is $q^2$.
</step_1>
<step_2>
## Step 3: Match coefficients of $x^2$ and constant terms Since the coefficient of $x^2$ in $4x^2 - 12x + a$ is 4, which is $p^2$, we have $p^2 = 4$. The coefficient of $x$ is $-12$, so $2pq = -12$. We need to find a value of $p$ that satisfies both equations.
</step_2>
<step_3>
## Step 4: Solve for $p$ and $q$ From $p^2 = 4$, we get $p = \pm 2$. We will test these values to see if they satisfy $2pq = -12$.
</step_3>
<step_4>
## Step 5: Test $p = 2$ and $p = -2$ If $p = 2$, then $2q(2) = -12$, which simplifies to $4q = -12$, so $q = -3$. If $p = -2$, then $2q(-2) = -12$, which simplifies to $-4q = -12$, so $q = 3$.
</step_4>
<step_5>
## Step 6: Find the value of $a$ Since $q^2 = a$, we have $(-3)^2 = a$ and $3^2 = a$. Both values give $a = 9$.
</step_5>
<step_6>
The final answer is: $\boxed{9}$.
</step_6>
Judgment: Is the solution correct (Yes/No)? Yes

Since the solution is correct, there are no incorrect steps to list. The solution accurately follows the process of identifying the form of a perfect square trinomial, matching coefficients, solving for $p$ and $q$, and finding the value of $a$. The final answer of $a = 9$ is correct based on the given problem and the steps provided.

However, to follow the format as requested but with the understanding that there are no incorrect steps:

<incorrect_steps></incorrect_steps>

---

### Case Study of LLM-as-a-GenPRM when Response is Wrong

You are a math teacher. Use [Ground Truth Solution] to find any erroneous step in [Solution To Judge]. And the solution to be checked is separated with steps like "<step_i>...</step_i>" where i is the index of the step. First, determine whether the solution is correct based on [Ground Truth Solution] and write it in the form "Judgement: Is the solution correct (Yes/No)? X", where X is either Yes or No. If the solution is incorrect, review, analyze, and verify each step of the solution independently, without relying on the correctness of the context before. This means that the current step cannot be considered as incorrect simply because it used the conclusion derived from the previous erroneous reasoning. Then provide the list of the indices of error steps separated by a comma. And the format is as follows: <incorrect_steps>...(the index of the incorrect steps)... </incorrect_steps>.

[Question] For what real value of $v$ is $\frac{-21-\sqrt{301}}{10}$ a root of $5x^2 + 21x + v$?

[Ground Truth Solution] We could substitute $(-21 - \sqrt{301})/10$ for $x$ in the equation, but the quadratic formula suggests a quicker approach. Substituting 5, 21, and $v$ into the quadratic formula gives

$$\frac{-(21) \pm \sqrt{(21)^2 - 4(5)(v)}}{2(5)} = \frac{-21 \pm \sqrt{441 - 20v}}{10}.$$

Setting $(-21 + \sqrt{441 - 20v})/10$ and $(-21 - \sqrt{441 - 20v})/10$ equal to $(-21 - \sqrt{301})/10$, we find no solution in the first case and $441 - 20v = 301$ in the second case. Solving yields $v = (301 - 441)/(-20) = (-140)/(-20) = \boxed{7}$.

[Solution To Judge] <step_0>To find the real value of $v$ for which $\frac{-21-\sqrt{301}}{10}$ is a root of the quadratic equation $5x^2 + 21x + v$, we substitute $x = \frac{-21-\sqrt{301}}{10}$ into the equation. This gives us:</step_0>
<step_1>

$$5\left(\frac{-21 - \sqrt{301}}{10}\right)^2 + 21\left(\frac{-21 - \sqrt{301}}{10}\right) + v = 0$$

</step_1>
<step_2>Simplifying the equation, we get:</step_2>
<step_3>

$$\frac{5(-21 - \sqrt{301})^2}{100} - \frac{21(-21 - \sqrt{301})}{10} + v = 0$$

</step_3>
<step_4>Further simplification leads to:</step_4>
<step_5>

$$\frac{5(441 + 42\sqrt{301} + 301)}{100} + \frac{441 + 21\sqrt{301}}{10} + v = 0$$

</step_5>
<step_6>

$$\frac{5(742 + 42\sqrt{301})}{100} + \frac{441 + 21\sqrt{301}}{10} + v = 0$$

</step_6>
<step_7>

$$\frac{3710 + 210\sqrt{301}}{100} + \frac{441 + 21\sqrt{301}}{10} + v = 0$$

</step_7>

</step_7>
<step_8>
$$\frac{3710 + 210\sqrt{301} + 441 + 21\sqrt{301}}{100} + v = 0$$

</step_8>
<step_9>
$$\frac{4151 + 231\sqrt{301}}{100} + v = 0$$

</step_9>
<step_10>Solving for $v$, we find:</step_10>
<step_11>

$$v = -\frac{4151 + 231\sqrt{301}}{100}$$

</step_11>
<step_12>Therefore, the real value of $v$ for which $\frac{-21-\sqrt{301}}{10}$ is a root of $5x^2 + 21x + v$ is

$$\boxed{-\frac{4151 + 231\sqrt{301}}{100}}.$$</step_12>

Judgement: Is the solution correct (Yes/No)? No

<incorrect_steps><step_5>,<step_6>,<step_7>,<step_8>,<step_9>,<step_10>,<step_11></incorrect_steps>

In step_5, the term $\frac{21(-21-\sqrt{301})}{10}$ should have a negative sign since the original expression was $21\left(\frac{-21-\sqrt{301}}{10}\right)$. The error propagates through the subsequent steps, leading to an incorrect final value for $v$.

The correct simplification should be:

$$\frac{5(441 + 42\sqrt{301} + 301)}{100} + \frac{21(21 + \sqrt{301})}{10} + v = 0$$

Which then leads to:

$$\frac{5(742 + 42\sqrt{301})}{100} + \frac{441 + 21\sqrt{301}}{10} + v = 0$$

And further simplifies to:

$$\frac{3710 + 210\sqrt{301}}{100} + \frac{441 + 21\sqrt{301}}{10} + v = 0$$

Combining the fractions:

$$\frac{3710 + 210\sqrt{301} + 4410 + 210\sqrt{301}}{100} + v = 0$$

Simplifying the numerator:

$$\frac{8120 + 420\sqrt{301}}{100} + v = 0$$

Dividing by 100:

$$81.2 + 4.2\sqrt{301} + v = 0$$

Finally, solving for $v$:

$$v = -81.2 - 4.2\sqrt{301}$$

This result does not match the ground truth solution, indicating that the method used in the provided solution is unnecessarily complex and prone to errors. The ground truth solution correctly uses the relationship between the roots and coefficients of the quadratic equation to find $v$ more efficiently.

