# OpenReview forum: "CAPO: Towards Enhancing LLM Reasoning through Generative Credit Assignment"
_ICLR.cc/2026/Conference — Submitted to ICLR 2026_

### Official Review · Reviewer_NbTj · 2025-10-31

**Soundness:** 2
**Presentation:** 2
**Contribution:** 2
**Rating:** 4
**Confidence:** 4

**Summary:**

This paper tackles the credit assignment problem in RLVR by leveraging an off-the-shelf LLM to judge step correctness. The resulting algorithm (CAPO) outperforms baselines in multiple base models and evaluation benchmarks.

**Strengths:**

1. The motivation of directly tackling credit assignment is clear, and discussion on related means are sound.
2. The discussion on design choices and ablations are abundant. The experiments are solid.

**Weaknesses:**

1. Querying an off-the-shielf LLM online is very costly and poses additional challenge to RLVR infrastructure. This limits the use of CAPO, and makes the comparison with baselines a bit unfair.
2. Lack of analysis on the motivation. Is CAPO assigning credit better? Does it lead to less error-prone generation?
3. The benchmark performances are not very strong, see questions below.

**Questions:**

1. Weakness 1. For each training token, how many tokens do LLM judges need to generate? If such additional cost are used in GRPO (e.g. more steps, longer contexts, larger group sizes), can CAPO still perform better?
2. Weakness 2. Is there evidence on CAPO learning better credit assignment, other than final benchmark scores? Can better feedback at train-time translate to inference-time (e.g. step error rate)?
3. Weakness 3. Let's focus on the largest scale Qwen-2.5-7B experiments.
- The most direct baseline, GRPO-Rule, reduces performance in almost all benchmarks. How to explain?
- The OOD general reasoning performance of CAPO generally decreases, while that of GRPO roughly maintains. How come better credit assignment hurts generalizability?
- The absolute performance is a bit low comparing to other works of similar setup. As an example, in [1], Qwen-2.5-7B with GRPO can improve score of Math500 (64.6 -> 78.2), AIME24 (0.3 -> 15.6) and AMC23 (30.0 -> 62.5), much higher than GRPO baseline in CAPO's report. This looks more strange given CAPO's experiment includes a cold-start SFT. Although different setups are not directly comparable, this large mismatch requires some explanation.

[1] SimpleRL-Zoo: Investigating and Taming Zero Reinforcement Learning for Open Base Models in the Wild

---

> ### Author Response · Authors · 2025-11-20
> **Rebuttal by Authors**
>
> Thank you for your insightful and careful review! We are glad that you found experiments solid. Below, we address the weaknesses you have raised:
>
> ***Note on Code Update: We have updated the supplementary materials and the anonymous link with a new implementation based on the widely adopted verl framework.*** This replaces the previous version, which relied on a proprietary internal RL infrastructure, to ensure better **accessibility and reproducibility for the community**.
>
> ### **W1: Infrastructure Challenges, Cost, and Fairness**
> **R1:** Thank you for these critical questions regarding the practicality and comparative fairness of using a Generative Reward Model. We address the infrastructure concerns and the cost-benefit analysis below, supported by new results.
>
> **1. Infrastructure Feasibility (W1.1)**
>
> Implementing CAPO does not impose significant infrastructure hurdles. The GenPRM is deployed as a decoupled service using vLLM on separate nodes. The use of generative reward models is also becoming common in modern RL pipelines [1] [2].
>
> **2. Fairness and "Compute-Matched" Comparison (W1.2)**
>
> **Our experiments suggest that given additional cost, GRPO would not perform better.** Following the standard Simple-RL [3] setting, we conducted extensive experiments across various sizes including 1B, 1.5B, 7B, and 14B models. We employed **Qwen2.5-14B-Instruct** for 1B, 1.5B and 7B models and **Qwen2.5-32B-Instruct** for 14B models as GenPRMs, rather than just 72B models. **More discussions of experiments setting please refer to response to W3.**
> * **GRPO Plateaus**: We analyzed the training dynamics in **Table 5**. GRPO converges rapidly (around step 90, one epoch) and hits a performance ceiling. Even when we extended training to **176 steps (doubling the compute budget)**, GRPO showed negligible improvement. This indicates that GRPO's limitation is not a lack of compute, but the sparsity of the reward signal.
> * **CAPO's Efficiency**: In contrast, CAPO overcomes this signal sparsity. CAPO consistently reaches higher accuracy than GRPO's converged baseline. CAPO also converge faster. In our Qwen2.5-Math-1.5B experiments (**Table 2**), using a stronger judge (32B) allowed CAPO to converge in just **1/3 of the steps** required by the baseline.
> In summary, simply giving GRPO more budget yields limited returns. CAPO invests that budget into better signal quality, yielding a higher return on compute.
>
> **3. Detailed GPU hours Cost Analysis**
>
> In our experiments, the judge typically generates fewer than 500 tokens per response to evaluate a full reasoning chain. We compare the GPU hours cost in training Qwen2.5-Math-7B for GRPO and CAPO with **a local 14B judge (the cost is also included in the calculation of the GPU hours)** : For the time per step, GRPO takes **~47s**; CAPO takes **~55s**. As for total GPU Hours: GRPO requires **272 hours** vs. CAPO's **331.2 hours** for a full run. Finally, CAPO introduces a **~21.7%** overhead. We argue this is **acceptable** given that it achieve the performance that GRPO cannot reach regardless of training duration. Furthermore, due to the faster convergence mentioned above, CAPO can often be early-stopped, potentially making the total GPU time comparable to or even less than training GRPO to convergence.
>
> **4. Reduced Dependence on Large Models**
>
> Finally, to address the cost of "off-the-shelf LLMs," we have expanded our experiments (**Tables 1-4**) to show that CAPO works effectively with smaller, open-source judges (e.g., Qwen2.5-14B-Instruct except for training 14B models using 32B-Instruct), removing the need for expensive 70B+ models while still outperforming GRPO.

---

> ### Author Response · Authors · 2025-11-20
> **Rebuttal by Authors (continued)**
>
> #### **Table 1: Performance Comparison across different methods using Llama3.2-1B-Instruct.** *CAPO uses Qwen2.5-14B-Instruct as a judge.*
> | model | gpqa | arc_c | mmlu_pro | OOD_Mean | math | olympiad_bench | amc | gsm8k | Math_Mean | All_Mean |
> |:---|:---:|:---:|:---:|:---:|:---:|:---:|:---:|:---:|:---:|:---:|
> | Base | 18.69 | 6.91 | 1.30 | 8.96 | 22.60 | 4.89 | 5.65 | 20.62 | 13.44 | 11.52 |
> | GRPO | 20.54 | 51.68 | 17.16 | 29.79 | 31.00 | **8.44** | 11.23 | 51.28 | 25.49 | 27.33 |
> | RLOO | **24.41** | 32.99 | 16.39 | 24.60 | 30.13 | 5.63 | 14.48 | 48.42 | 24.67 | 24.64 |
> | PRM | 14.98 | 52.59 | 16.58 | 28.05 | 31.87 | 7.01 | 12.58 | 51.07 | 25.63 | 26.67 |
> | CAPO | 17.60 | **54.78** | **17.56** | **29.98** | **33.13** | 8.30 | **17.41** | **51.55** | **27.60** | **28.62** |
>
> ---
>
> #### **Table 2: Performance comparison across different methods using Qwen2.5-Math-1.5B.** *CAPO uses Qwen2.5-14B-Instruct as a judge.*
> | model | gpqa | arc_c | mmlu_pro | OOD_Mean | math | olympiad_bench | minerva | aime | amc | Math_Mean | All_Mean |
> |:---|:---:|:---:|:---:|:---:|:---:|:---:|:---:|:---:|:---:|:---:|:---:|
> | Base | 5.05 | 18.20 | 10.30 | 11.18 | 63.13 | 29.43 | 12.25 | 9.97 | 41.57 | 31.27 | 23.74 |
> | GRPO | **14.65** | **50.09** | 23.48 | **29.40** | 76.40 | 38.32 | 31.62 | 6.67 | 42.31 | 39.06 | 35.44 |
> | RLOO | 13.47 | 39.11 | 21.60 | 24.73 | 75.60 | 35.70 | 29.66 | 6.67 | 49.75 | 39.48 | 33.94 |
> | PRM | 10.61 | 45.88 | 22.99 | 26.49 | 76.00 | 38.81 | **32.35** | 12.40 | 43.30 | 40.57 | 35.29 |
> | CAPO | 13.13 | 43.34 | **24.28** | 26.92 | **76.80** | **40.20** | 32.23 | 18.19 | 44.73 | 42.43 | **36.61** |
> | CAPO_GenRM32B 1/3 steps | 13.64 | 41.81 | 20.46 | 25.30 | 74.70 | 38.00 | 28.86 | **23.18** | **49.85** | **42.92** | 36.31 |
>
> ---
>
> #### **Table 3: Performance comparison across different methods using Qwen2.5-Math-7B.** *CAPO uses Qwen2.5-14B-Instruct as a judge.*
> | model | gpqa | arc_c | mmlu_pro | OOD_Mean | math | olympiad_bench | minerva | aime | amc | Math_Mean | All_Mean |
> |:---|:---:|:---:|:---:|:---:|:---:|:---:|:---:|:---:|:---:|:---:|:---:|
> | Base | 18.18 | 30.32 | 16.71 | 21.74 | 64.73 | 19.80 | 9.07 | 11.35 | 42.28 | 29.45 | 26.56 |
> | GRPO | **25.76** | 64.11 | **44.62** | **44.83** | 81.13 | 44.10 | 36.76 | 30.00 | 56.78 | 49.75 | 47.91 |
> | RLOO | 22.22 | 42.43 | 41.84 | 35.50 | 79.33 | 41.73 | 33.95 | **40.52** | 59.46 | 51.00 | 45.19 |
> | PRM | 20.03 | **70.73** | 42.08 | 44.28 | 81.60 | 43.11 | **38.36** | 24.34 | **61.02** | 49.69 | 47.66 |
> | CAPO | 23.89 | 65.85 | 43.20 | 44.31 | **81.67** | **45.14** | 38.11 | 35.00 | 59.10 | **51.80** | **48.99** |
>
> ---
>
> #### **Table 4: Performance comparison across different methods using Qwen2.5-14B.** *CAPO uses Qwen2.5-32B-Instruct as a judge.*
> | model | gpqa | arc_c | mmlu_pro | OOD_Mean | math | olympiad_bench | minerva | aime24 | amc | aime25 | Math_Mean | All_Mean |
> |:---|:---:|:---:|:---:|:---:|:---:|:---:|:---:|:---:|:---:|:---:|:---:|:---:|
> | Base | 25.76 | 76.54 | 54.89 | 52.39 | 61.60 | 29.19 | 25.37 | 10.00 | 33.85 | 5.21 | 27.53 | 35.82 |
> | GRPO | 30.30 | 86.18 | 61.96 | 59.48 | **81.40** | **44.59** | 38.60 | 16.67 | 50.45 | 5.63 | 39.56 | 46.20 |
> | CAPO | **30.81** | **90.19** | **62.06** | **61.02** | 80.00 | 42.96 | **41.91** | **22.81** | **52.97** | **13.44** | **42.35** | **48.57** |
>
> ---
>
> #### **Table 5: Performance comparison of different steps using Qwen2.5-Math-7B.** *CAPO uses Qwen2.5-14B-Instruct as a judge.*
> | model_name | step | gpqa | arc_c | mmlu_pro | OOD_Mean | math | olympiad_bench | minerva | aime | amc | Math_Mean | All_Mean |
> |:---|:---:|:---:|:---:|:---:|:---:|:---:|:---:|:---:|:---:|:---:|:---:|:---:|
> | Base | | 18.18 | 30.32 | 16.71 | 21.74 | 64.73 | 19.80 | 9.07 | 11.35 | 42.28 | 29.45 | 26.56 |
> | GRPO | 60 | 23.23 | 59.44 | 43.69 | 42.12 | 80.53 | 41.53 | 38.73 | 26.67 | 61.38 | 49.77 | 46.90 |
> | | 90 | 22.90 | 65.22 | 44.70 | 44.27 | 79.33 | 40.89 | 38.85 | 28.96 | 50.62 | 47.73 | 46.43 |
> | | 120 | 24.75 | 60.75 | 44.69 | 43.40 | 81.93 | 43.56 | 34.56 | 24.17 | 60.17 | 48.88 | 46.82 |
> | | 176 | 25.76 | 64.11 | 44.62 | 44.83 | 81.13 | 44.10 | 36.76 | 30.00 | 56.78 | 49.75 | 47.91 |
> | CAPO | 60 | 22.89 | 58.06 | 43.94 | 41.63 | 81.33 | 43.17 | 38.03 | 28.11 | 60.49 | 50.23 | 47.00 |
> | | 90 | 20.01 | 60.89 | 43.94 | 41.61 | 82.27 | 43.11 | 38.97 | 30.00 | 60.99 | 51.07 | 47.52 |
> | | 120 | 23.89 | 65.85 | 43.20 | 44.31 | 81.67 | 45.14 | 38.11 | 35.00 | 59.10 | 51.80 | 48.99 |
> | | 176 | 23.74 | 69.23 | 39.57 | 44.18 | 80.33 | 40.30 | 37.38 | 26.56 | 50.94 | 47.10 | 46.01 |

---

> ### Author Response · Authors · 2025-11-20
> **Rebuttal by Authors (continued)**
>
> ### **W2: Validation of the Motivation to Improve Reasoning Process Quality**
> **R2:** Thank you for this critical comment. We agree that demonstrating an actual improvement in the quality of the reasoning process—beyond just final accuracy—is essential to support our core hypothesis. We have added a comprehensive process-level error analysis.
>
> Conducting a process judge on full test set (spanning 19,425 samples across Math and OOD benchmarks, with contexts exceeding 10k tokens) is prohibitively expensive using outer APIs. We employed Qwen2.5-72B-Instruct as a highly capable judge to inspect the reasoning paths generated by the models trained in experiments (**specifically, the GRPO and CAPO models from Tables 5-7 are trained using the Simple-RL[6] setting, and with a 14B judge for CAPO instead of 72B in the original paper**).
>
> To rigorously quantify the quality of the reasoning process, we define two metrics: **Step Error Ratio (SER)**: The percentage of reasoning steps identified as incorrect and **Token Error Ratio(TER)**: The percentage of total tokens belonging to incorrect steps.
> Let dataset $\mathcal{D}$ contain $N$ generated responses. For the $i$-th response $Y_i$, let it consist of $K_i$ steps $(s_{i,1}, \dots, s_{i,K_i})$, where $|s_{i,k}|$ denotes the token count of step $s_{i,k}$, and $\mathbb{I}(s_{i,k})$ is the error indicator ($1$ if incorrect, $0$ otherwise). We calculate the metrics by averaging the error rates per response:
>
> \begin{equation}
> \text{SER} = \frac{1}{N} \sum_{i=1}^{N} \left( \frac{\sum_{k=1}^{K_i} \mathbb{I}(s_{i,k})}{K_i} \right)
> \end{equation}
>
> \begin{equation}
> \text{TER} = \frac{1}{N} \sum_{i=1}^{N} \left( \frac{\sum_{k=1}^{K_i} \mathbb{I}(s_{i,k}) \cdot |s_{i,k}|}{\sum_{k=1}^{K_i} |s_{i,k}|} \right)
> \end{equation}
>
>  This macro-averaging approach ensures that short and long reasoning chains are weighted equally, providing an unbiased evaluation of model performance across varying problem complexities.
>
>
> As shown in the **Table5-7** below, CAPO demonstrates a **significant reduction in process errors compared to GRPO** across both In-Domain (Math) and Out-Of-Domain benchmarks and achieve an average reduce up to 7%. The lower Error Token Ratio confirms that CAPO generates "cleaner" reasoning chains with fewer logic or calculation errors. These findings directly validate the motivation of CAPO. By providing reliable, fine-grained credit assignment, CAPO **effectively penalizes specific erroneous steps during training**. Unlike GRPO, which may reward a correct answer derived from flawed reasoning, CAPO forces the model to attend to the correctness of each intermediate step, thereby successfully shifting the learning objective from purely outcome-oriented to process-oriented.
>
> ---
>
> #### **Table 5: Comparison of step error ratio and token error ratio across different methods using Llama3.2-1B-Instruct**
> | Model | aime (SER) | aime (TER) | amc (SER) | amc (TER) | arc_c (SER) | arc_c (TER) | gpqa (SER) | gpqa (TER) | math (SER) | math (TER) | minerva (SER) | minerva (TER) | mmlu_pro (SER) | mmlu_pro (TER) | olympiad (SER) | olympiad (TER) |  OOD_Mean (SER) |  OOD_Mean (TER) | Math_Mean (SER) | Math_Mean (TER) |
> | :--- | :---: | :---: | :---: | :---: | :---: | :---: | :---: | :---: | :---: | :---: | :---: | :---: | :---: | :---: | :---: | :---: | :---: | :---: | :---: | :---: |
> | GRPO | 67.29 | 67.66 | 52.06 | 52.83 | 25.22 | 22.80 | 55.65 | 59.41 | 40.31 | 38.40 | 56.56 | 57.73 | 47.33 | 45.30 | 56.49 | 56.63 | 42.73 | 42.50 | 54.54 | 54.65 |
> | CAPO | **51.10** | **50.41** | **49.01** | **48.76** | **23.06** | **21.19** | **49.02** | **49.43** | **37.95** | **34.71** | **55.54** | **54.46** | **45.73** | **41.59** | **51.72** | **49.94** | **39.27** | **37.40** | **49.06** | **47.66** |
>
> ---
>
> #### **Table 6: Comparison of step error ratio and token error ratio across different methods using Qwen2.5-Math-1.5B**
> | Model | aime (SER) | aime (TER) | amc (SER) | amc (TER) | arc_c (SER) | arc_c (TER) | gpqa (SER) | gpqa (TER) | math (SER) | math (TER) | minerva (SER) | minerva (TER) | mmlu_pro (SER) | mmlu_pro (TER) | olympiad (SER) | olympiad (TER) |  OOD_Mean (SER) |  OOD_Mean (TER) | Math_Mean (SER) | Math_Mean (TER) |
> | :--- | :---: | :---: | :---: | :---: | :---: | :---: | :---: | :---: | :---: | :---: | :---: | :---: | :---: | :---: | :---: | :---: | :---: | :---: | :---: | :---: |
> | GRPO | 27.62 | 41.11 | 19.64 | 23.99 | 23.66 | 23.74 | 37.74 | 48.01 | 10.93 | 11.12 | 19.96 | 24.94 | 33.43 | 33.03 | 22.62 | 26.34 | 31.61 | 34.93 | 20.15 | 25.50 |
> | CAPO | **26.01** | **34.93** | **13.01** | **14.87** | **20.60** | **20.16** | **36.31** | **44.55** | **7.45** | **8.06** | **15.52** | **19.18** | **27.41** | **28.32** | **17.79** | **18.39** | **28.11** | **31.01** | **15.96** | **19.09** |

---

> ### Author Response · Authors · 2025-11-20
> **Rebuttal by Authors (continued)**
>
> #### **Table 7: Comparison of step error ratio and token error ratio across different methods using Qwen2.5-Math-7B**
> | Model | aime (SER) | aime (TER) | amc (SER) | amc (TER) | arc_c (SER) | arc_c (TER) | gpqa (SER) | gpqa (TER) | math (SER) | math (TER) | minerva (SER) | minerva (TER) | mmlu_pro (SER) | mmlu_pro (TER) | olympiad (SER) | olympiad (TER) |  OOD_Mean (SER) |  OOD_Mean (TER) | Math_Mean (SER) | Math_Mean (TER) |
> | :--- | :---: | :---: | :---: | :---: | :---: | :---: | :---: | :---: | :---: | :---: | :---: | :---: | :---: | :---: | :---: | :---: | :---: | :---: | :---: | :---: |
> | GRPO | 32.83 | 34.37 | 12.31 | 14.82 | 12.94 | 12.23 | 29.55 | 34.01 | **4.96** | **5.49** | 15.10 | 15.84 | 30.86 | 30.09 | 15.24 | 16.47 | 24.45 | 25.44 | 16.09 | 17.40 |
> | CAPO | **20.16** | **23.17** | **10.46** | **12.38** | **7.44** | **6.42** | **24.76** | **30.50** | 5.98 | 5.82 | **10.72** | **13.15** | **20.29** | **18.94** | **13.93** | **16.03** | **17.49** | **18.62** | **12.25** | **14.11** |
>
> ---
>
> ### **W3 & Q3: Analysis of Baselines, Generalization, and Absolute Performance**
> **R3:** Thank you for the rigorous examination of our experimental results! We fully acknowledge the discrepancies you pointed out: the suboptimal performance of the GRPO baseline in our initial 7B experiments and the gap compared to benchmarks like Simple-RL [3].
>
> **1.Cause Analysis: The "Cold-Start SFT" Issue**
>
> We attribute the lower absolute performance and the instability of GRPO in our initial draft to the specific "Cold-Start SFT followed by RL" setup. Our analysis suggests the SFT phase was overly aggressive (too many steps/epochs), causing the model to overfit to a narrow distribution [4]. This "mode collapse" severely limited the exploration capability of the policy during the subsequent RL phase, causing methods like GRPO to degrade performance rather than improve it. The mismatches in data distribution during SFT also likely hindered the performance during RL.
>
> **2. New Experimental Setup aligning with Simple-RL (Zero-RL)**
>
> To address this and provide a fair, direct comparison with the community benchmark, **we have completely re-executed our experiments** following the widely adopted direct RL on base models (**Zero-RL**) setting, as formalized in Simple-RL [3].
> * Setup: We utilized the standardized 8k difficulty-stratified dataset from [3] and applied RL directly to base models (Qwen2.5-Math-1.5B/7B, Qwen2.5-14B, and Llama3.2-1B-Instruct). For CAPO, we used the smaller Qwen2.5-14B-Instruct as the GenPRM (instead of 72B) and 32B for 14B base model.
> * Baselines: We added robust baselines, including RLOO[6], PRM-based credit assignment [5], and GRPO.
>
> The new results are presented in **Tables 5, 6, 7, 8 and 9**. Our new baseline results now align close with those reported in Simple-RL [3], though some trade-offs existed like AIME and AMC.  In this setup, CAPO consistently outperforms GRPO, RLOO and PRM-based approaches in math tasks and leads in overall performances, without hurting the general reasoning.
>
> **3. CAPO vs. Traditional PRMs**
>
> A noteworthy finding from our new Llama-1B experiments is that PRM-based baselines performed poorly, even worse than GRPO. We hypothesize this is due to **distribution shift**: open-source PRMs are typically trained on Qwen/DeepSeek data, making them less effective judges for Llama. CAPO, which relies on a general-purpose instruction-following judge (GenPRM) with Ground Truth, proved much more robust to this architecture shift, achieving superior performance.
> By aligning with the standard Simple-RL setting, we have demonstrated that the initial performance drop was an artifact of the SFT setup, not a flaw in CAPO. We hope these results could resolve your doubts.
>
> ---
>
> [1] Chen, Xiusi, et al. "Rm-r1: Reward modeling as reasoning." arXiv 2025.
>
> [2] Ding, Yuyang, et al. "FAPO: Flawed-Aware Policy Optimization for Efficient and Reliable Reasoning." arXiv 2025
>
> [3] Zeng, Weihao, et al. "Simplerl-zoo: Investigating and taming zero reinforcement learning for open base models in the wild." COLM 2025.
>
> [4] Chu, Tianzhe, et al. "Sft memorizes, rl generalizes: A comparative study of foundation model post-training." ICML2025.
>
> [5] Cheng, Jie, et al. "Stop summation: Min-form credit assignment is all process reward model needs for reasoning." NeurIPS 2025
>
> [6] Ahmadian, Arash, et al. "Back to basics: Revisiting reinforce style optimization for learning from human feedback in llms." ACL2024

---

### Official Review · Reviewer_zJU4 · 2025-11-01

**Soundness:** 3
**Presentation:** 3
**Contribution:** 3
**Rating:** 6
**Confidence:** 2

**Summary:**

The paper introduces Credit Assignment Policy Optimization (CAPO), an efficient and simple methodology designed to enhance Large Language Model (LLM) reasoning by resolving the coarse-grained credit assignment issue inherent in existing Reinforcement Learning with Verifiable Rewards (RLVR) methods. CAPO avoids training auxiliary models by leveraging an off-the-shelf, general-purpose LLM as a Generative Process Reward Model (LLM-as-GenPRM), which efficiently generates all step-wise critiques in a single pass based on the intrinsic correctness of each step. This provides deterministic, token-level credits to refine the original coarse rewards, enabling a crucial shift from purely outcome-oriented to process-oriented learning. The approach incorporates voting mechanisms for enhanced robustness and utilizes a structured reward design to effectively balance outcome and process objectives. Extensive experiments on various LLM backbones demonstrate that CAPO consistently outperforms supervised learning and other RL-based fine-tuning methods across multiple mathematical and out-of-domain reasoning benchmarks, proving its ability to foster the learning of correct reasoning pathways.

**Strengths:**

1. CAPO is "elegantly simple" and avoids the need for complex, time-consuming auxiliary models (like PRMs or value models in PPO) or costly high-quality process supervision data. It efficiently generates all step-wise credit in a single pass.

2. By prompting a powerful LLM to focus on the intrinsic correctness of each step, CAPO provides deterministic process credits, which are more reliable and less susceptible to reward hacking compared to probabilistic, estimation-based signals.

3. The framework utilizes off-the-shelf, powerful LLMs as the GenPRM, making the method broadly applicable, easily reproducible, and accessible since it does not require specialized or finely-tuned reward models. The GenPRM can even be accessed via APIs to minimize local computational footprint.

4. The method consistently achieves superior performance over strong baselines (SFT, GRPO) on both in-domain mathematical and out-of-domain general reasoning tasks across multiple LLM scales. Analysis confirms it actively fosters the learning of correct reasoning pathways.

**Weaknesses:**

1. The efficacy and reliability of the process credit mechanism are predicated on the use of an LLM-as-GenPRM that is significantly more capable than the policy model being trained (e.g., using a 70B+ model to guide a 1B-7B model). This relies on the availability of a superior external model.

2. The introduction of process-level rewards creates a potential conflict with the original outcome-level reward signal. This required an in-depth analysis and the proposal of a non-trivial asymmetric reward shaping strategy to create a functional hierarchical learning objective.

3. The current credit assignment is performed at the step level, typically delineated by \n \n. While practical, this step-level granularity is not truly token-level and its reliability may depend on the consistency of the policy model's generation format.

**Questions:**

none

---

> ### Author Response · Authors · 2025-11-20
> **Rebuttal by Authors**
>
> Thank you for your insightful and positive review! We are glad that you found method to be "elegantly simple", superior and with broad applicability and accessibility!
>
> Below, we address the weaknesses you have raised:
>
>
> ***Note on Code Update: We have updated the supplementary materials and the anonymous link with a new implementation based on the widely adopted verl framework.*** This replaces the previous version, which relied on a proprietary internal RL infrastructure, to ensure better **accessibility and reproducibility for the community**.
>
>
> ### **W1: On the Dependency on a More Capable GenPRM**
> **R1:** Thank you for raising this important point regarding the scalability of our approach and the "weak-to-strong" generalization gap. While we agree that all guidance-based paradigms (whether PRMs, Critics, or Teachers) benefit from stronger models, we offer three key arguments supported by new experiments to address the concern about the "scalability ceiling."
>
> **1. Reducing Dependency via Ground Truth (Verification is Easier than Generation)**
>
> A critical design choice in CAPO effectively lowers the capability requirement for the GenPRM: access to the Ground Truth. Unlike a standard solver that must reason from scratch, the GenPRM in CAPO is conditioned on the correct final answer during the critique phase. This shifts the task from generation to verification which is easier than generating it from scratch. Therefore, we don't need a significantly strong model to be a judge.
>
> **2. Empirical Evidence with Smaller Judges**
>
> As also seen in response to W1, we conducted extensive new experiments using Qwen2.5-14B-Instruct as the GenPRM and increase more baselines including other RLVR approaches RLOO [7] and Process Reward Models based credit assignment approaches [6]. As shown in Tables 1, 2, 3, and 4, CAPO driven by a 14B judge consistently outperforms baselines and achieves gains comparable to those obtained with a 72B judge.
>
> **3. Accessibility**
>
> Finally, we highlight a practical advantage. Traditional PRM-based methods often require expensive additional training (costly data collection + model fine-tuning) to create a capable verifier. In contrast, CAPO leverages off-the-shelf, general-purpose models (even smaller ones like 14B) without any additional training. This makes the method highly accessible and scalable, as it bypasses the need to train a "super-judge" from scratch.
>
> ---
>
> #### **Table 1: Performance Comparison across different methods using Llama3.2-1B-Instruct**
> | model | gpqa | arc_c | mmlu_pro | OOD_Mean | math | olympiad_bench | amc | gsm8k | Math_Mean | All_Mean |
> |:---|:---:|:---:|:---:|:---:|:---:|:---:|:---:|:---:|:---:|:---:|
> | Base | 18.69 | 6.91 | 1.30 | 8.96 | 22.60 | 4.89 | 5.65 | 20.62 | 13.44 | 11.52 |
> | GRPO | 20.54 | 51.68 | 17.16 | 29.79 | 31.00 | **8.44** | 11.23 | 51.28 | 25.49 | 27.33 |
> | RLOO | **24.41** | 32.99 | 16.39 | 24.60 | 30.13 | 5.63 | 14.48 | 48.42 | 24.67 | 24.64 |
> | PRM | 14.98 | 52.59 | 16.58 | 28.05 | 31.87 | 7.01 | 12.58 | 51.07 | 25.63 | 26.67 |
> | CAPO | 17.60 | **54.78** | **17.56** | **29.98** | **33.13** | 8.30 | **17.41** | **51.55** | **27.60** | **28.62** |
>
> ---
>
> #### **Table 2: Performance comparison across different methods using Qwen2.5-Math-1.5B**
> | model | gpqa | arc_c | mmlu_pro | OOD_Mean | math | olympiad_bench | minerva | aime | amc | Math_Mean | All_Mean |
> |:---|:---:|:---:|:---:|:---:|:---:|:---:|:---:|:---:|:---:|:---:|:---:|
> | Base | 5.05 | 18.20 | 10.30 | 11.18 | 63.13 | 29.43 | 12.25 | 9.97 | 41.57 | 31.27 | 23.74 |
> | GRPO | **14.65** | **50.09** | 23.48 | **29.40** | 76.40 | 38.32 | 31.62 | 6.67 | 42.31 | 39.06 | 35.44 |
> | RLOO | 13.47 | 39.11 | 21.60 | 24.73 | 75.60 | 35.70 | 29.66 | 6.67 | 49.75 | 39.48 | 33.94 |
> | PRM | 10.61 | 45.88 | 22.99 | 26.49 | 76.00 | 38.81 | **32.35** | 12.40 | 43.30 | 40.57 | 35.29 |
> | CAPO | 13.13 | 43.34 | **24.28** | 26.92 | **76.80** | **40.20** | 32.23 | 18.19 | 44.73 | 42.43 | **36.61** |
> | CAPO_GenRM32B 1/3 steps | 13.64 | 41.81 | 20.46 | 25.30 | 74.70 | 38.00 | 28.86 | **23.18** | **49.85** | **42.92** | 36.31 |
>
>
>
> #### **Table 3: Performance comparison across different methods using Qwen2.5-Math-7B**
> | model | gpqa | arc_c | mmlu_pro | OOD_Mean | math | olympiad_bench | minerva | aime | amc | Math_Mean | All_Mean |
> |:---|:---:|:---:|:---:|:---:|:---:|:---:|:---:|:---:|:---:|:---:|:---:|
> | Base | 18.18 | 30.32 | 16.71 | 21.74 | 64.73 | 19.80 | 9.07 | 11.35 | 42.28 | 29.45 | 26.56 |
> | GRPO | **25.76** | 64.11 | **44.62** | **44.83** | 81.13 | 44.10 | 36.76 | 30.00 | 56.78 | 49.75 | 47.91 |
> | RLOO | 22.22 | 42.43 | 41.84 | 35.50 | 79.33 | 41.73 | 33.95 | **40.52** | 59.46 | 51.00 | 45.19 |
> | PRM | 20.03 | **70.73** | 42.08 | 44.28 | 81.60 | 43.11 | **38.36** | 24.34 | **61.02** | 49.69 | 47.66 |
> | CAPO | 23.89 | 65.85 | 43.20 | 44.31 | **81.67** | **45.14** | 38.11 | 35.00 | 59.10 | **51.80** | **48.99** |

---

> > ### Author Response · Authors · 2025-11-20
> > **Rebuttal by Authors (continued)**
> >
> > #### **Table 4: Performance comparison across different methods using Qwen2.5-14B**
> > | model | gpqa | arc_c | mmlu_pro | OOD_Mean | math | olympiad_bench | minerva | aime24 | amc | aime25 | Math_Mean | All_Mean |
> > |:---|:---:|:---:|:---:|:---:|:---:|:---:|:---:|:---:|:---:|:---:|:---:|:---:|
> > | Base | 25.76 | 76.54 | 54.89 | 52.39 | 61.60 | 29.19 | 25.37 | 10.00 | 33.85 | 5.21 | 27.53 | 35.82 |
> > | GRPO | 30.30 | 86.18 | 61.96 | 59.48 | **81.40** | **44.59** | 38.60 | 16.67 | 50.45 | 5.63 | 39.56 | 46.20 |
> > | CAPO | **30.81** | **90.19** | **62.06** | **61.02** | 80.00 | 42.96 | **41.91** | **22.81** | **52.97** | **13.44** | **42.35** | **48.57** |
> >
> > ---
> >
> > ### **W2: Rationale for the Asymmetric Reward Shaping Strategy**
> > **R2:** Thank you for this insightful comment. We agree that managing the interplay between process-level and outcome-level rewards is a critical challenge, and we have already dedicated a significant portion of our work to analyzing this dynamic.
> >
> > Due to space constraints in the main paper, we summarized our core insights on this interplay in Section 3.2. As we further elaborate in Section 4.2 and Appendix E, our investigation revealed that the two signals serve distinct but complementary roles at different phases of training:
> > * Early Stages: The outcome reward dominates. We found that the outcome-based reward is crucial for rapid initial learning. An overemphasis on the process signal at this stage is counterproductive, as a model with limited capabilities may learn to generate verbose but incorrect steps, a finding supported by our analysis in Figure 4 in the paper.
> > * Later Stages: The process reward becomes critical. Once the model consistently achieves correct outcomes, the training can plateau. At this point, the process signal is vital. It helps the model differentiate between solutions that are merely correct and those that are reasoned elegantly, thereby fostering more robust learning.
> > Therefore, our proposed asymmetric reward shaping is not a casual choice but a deliberate strategy derived directly from this analysis. It is explicitly designed to leverage the strengths of both signals at the appropriate times. The success of this approach is empirically validated in our ablation study (Section 4.2) and Appendix E, which confirms that our hierarchical objective effectively resolves this conflict.
> > We believe this in-depth analysis and the resulting reward framework are a core strength of our paper.
> >
> > ---
> >
> > ### **W3: Step-Level vs. Token-Level Credit Assignment**
> > **R3:** Thank you for this insightful comment. Our method indeed operates at the step-level granularity (assigning credit to the segment ending in \n\n), rather than estimating a value for every single token. We clarify our design choices and the adaptability of the framework below.
> >
> > **1. The Strategic Trade-off: Granularity vs. Reliability**
> >
> > We argue that "step-level" assignment represents a trade-off between the outcome-only rewards (RLVR) and fully dense token-level value estimation (like PPO). While PPO's value model provides a signal for tokens, estimating the contribution of each token is inherently noisy. This often leads to severe reward hacking[4] [5]. We sacrifice the "theoretical" granularity of per-token valuation for the "practical" robustness of step-wise verification. This design mitigates reward hacking while still offering significantly richer signals than sparse outcome rewards, effectively accelerating convergence and fostering better reasoning chains.
> >
> > **2. Community standards of \n\n in Math Reasoning**
> >
> > The choice using \n\n aligns with the community standards for math task, particularly for the Qwen model family and Process Reward Model literature [1][2].  It is also important to note that CAPO is a framework agnostic to the specific delimiter. The \n\n token is simply for math domain. For other tasks, the segmentation definition can be easily adapted. For instance, in agentic workflows, a "step" would be defined by a "turn of interaction". While adaptive or learnable segmentation [3] is a future direction, our focus is on establishing a simple, efficient credit assignment mechanism within the prevailing paradigms.
> >
> > ---
> >
> > [1] Zhang, Zhenru, et al. "The lessons of developing process reward models in mathematical reasoning." ACL2025
> >
> > [2] Zheng, Chujie, et al. "Processbench: Identifying process errors in mathematical reasoning." ACL2025
> >
> > [3] Liu, Yuliang, et al. "Adaptivestep: Automatically dividing reasoning step through model confidence." arXiv 2025
> >
> > [4] Gao, Leo, John Schulman, and Jacob Hilton. "Scaling laws for reward model overoptimization." ICML2023
> >
> > [5] Chen, Lichang, et al. "Odin: Disentangled reward mitigates hacking in rlhf." ICML 2024
> >
> > [6] Cheng, Jie, et al. "Stop summation: Min-form credit assignment is all process reward model needs for reasoning." NeurIPS 2025
> >
> > [7] Ahmadian, Arash, et al. "Back to basics: Revisiting reinforce style optimization for learning from human feedback in llms." ACL2024

---

> > > ### Comment · Reviewer_zJU4 · 2025-11-20
> > >
> > > Thank you for your reply. I believe my questions have been resolved, so I have decided to increase my rating.

---

> > > > ### Author Response · Authors · 2025-11-21
> > > > **Please let us know if you have any further questions**
> > > >
> > > > Thank you for your response, and we are glad that we addressed your questions! Please let us know if you have any questions since this would help us improve the work. Finally, we sincerely appreciate you raising the score.

---

### Official Review · Reviewer_RfL9 · 2025-11-01

**Soundness:** 3
**Presentation:** 4
**Contribution:** 3
**Rating:** 4
**Confidence:** 4

**Summary:**

This paper introduces CAPO (or the proposed method's name), a novel approach to enhance LLM reasoning through a stepwise PRM. The key innovation lies in moving beyond traditional, coarse-grained reward methods by assigning differential rewards to tokens based on the specific reasoning step they belong to. This fine-grained credit assignment is intended to better pinpoint which parts of the reasoning chain contribute to success or failure. The method demonstrates some empirical improvement on selected reasoning benchmarks and includes a valuable ablation study on the reward weighting.

**Strengths:**

1. Methodological Innovation: The method proposes an innovative approach using an LLM to implement a stepwise PRM. This design provides a more granular form of credit assignment by attributing different reward values to tokens depending on their corresponding reasoning step, which is a significant step beyond whole-response-based RL methods.

2. The paper includes a valuable ablation study on the different weights of the $P$ term (as presented in Table 4). This analysis effectively investigates the model's sensitivity to this key hyperparameter and provides insight into the optimal balance between the different reward components.

3. Empirical Results: The approach demonstrates a certain degree of improvement on reasoning tasks, suggesting the potential of a stepwise credit assignment mechanism.

**Weaknesses:**

1. The definition and segmentation of the output reasoning steps is a fundamental and critical aspect of this method. However, the paper only briefly addresses this matter in Appendix D. There is no sufficient justification for the forced output mechanism using markers like <step k>. The reader is left to wonder if a more principled or adaptive step segmentation strategy could be devised to yield a significantly better-performing PRM.

2. The method only achieves a not entirely convincing improvement when tested on models with fewer than 10B parameters. This restricts the ability to generalize the findings and claim the method's effectiveness across more LLMs.

3. A key motivation of this work is to improve the quality of the reasoning process. Despite this, the experimental section lacks a qualitative analysis that demonstrates an actual improvement in the quality of the process outputs, which is necessary to support the paper's core hypothesis.

4. Typos/Minor Issues: There are some minor editorial issues, such as the 'r' on line 189, which should be capitalized (i.e., 'R').

**Questions:**

In Equation 2, the reward term $R_{i,t}$ undergoes group normalization by subtracting the mean of $R$ across all tokens in all other rollouts. Given that the Process Reward Model is designed to assign a reward specific to the current step $i$ and token $t$, is this form of group normalization appropriate and theoretically sound for a step-wise reward model? Could this global normalization potentially obscure the fine-grained, step-specific signal that the method aims to capture?

---

> ### Author Response · Authors · 2025-11-20
> **Rebuttal by Authors**
>
> Thank you for your detailed and careful comment! We are glad that you found our CAPO approach novel and the ablation study about reward weight valuable and insightful!
>
> Below, we address the weaknesses you have raised:
>
> ***Note on Code Update: We have updated the supplementary materials and the anonymous link with a new implementation based on the widely adopted verl framework.*** This replaces the previous version, which relied on a proprietary internal RL infrastructure, to ensure better **accessibility and reproducibility for the community**.
>
> ### **W1: Clarity on the Definition and Segmentation of Reasoning Steps**
> **R1:** Thank you for your valuable comment. We agree that step segmentation is a foundational component of our method, and we apologize that the detailed discussion was previously relegated to Appendix D due to space constraints. In the revision, we will move the definition and justification of step segmentation to the main text to ensure clarity. We address your specific concerns below:
>
> **1. Justification for the \n\n Delimiter**
>
> The choice of \n\n as a step delimiter is not arbitrary but follows the widely accepted convention in the Process Reward Model (PRM) literature [1] [2] [3]. Using this delimiter ensures that our forced output mechanism aligns with the model's native distribution, minimizing inference artifacts.
>
> **2. Adaptability of CAPO to Other Domains**
>
> It is important to clarify that CAPO is a general framework for credit assignment, and the use of \n\n is merely **a domain-specific implementation** for mathematical reasoning. CAPO is agnostic to how steps are defined and can be easily adapted to other scenarios. In an Agentic workflow, the natural segmentation might be a "turn of interaction" or an "API call" rather than a newline character. CAPO can be directly applied to these segments without modification to the core algorithm.
>
> While we acknowledge that adaptive or learnable step segmentation [4] is a promising direction that could potentially yield better performance, our work focuses on introducing **a simpler and more efficient credit assignment mechanism** given a defined reasoning structure. We view adaptive segmentation as an orthogonal research line that could be integrated with CAPO in the future. We appreciate your suggestion and will highlight this as a valuable direction for future work in the revision.
>
> ---
>
>
>
> ### **W2: On the Generalization across Model Scales**
>
> **R2:** Thank you for the valuable comment. We understand your concern about the generalization of our findings, specifically whether the effectiveness of CAPO is limited when applied to different model scales. To address this, we extended our experiments beyond 10B and also significantly expanded our experiments using more efficient models as judges **instead of 72B in our original paper**, and including more baselines.
>
> * **Experimental Setup**: Following the standard Simple-RL [6] setting, we conducted extensive experiments across various sizes including 1B, 1.5B, 7B, and 14B models. We employed **Qwen2.5-14B-Instruct** for 1B, 1.5B and 7B models and **Qwen2.5-32B-Instruct** for 14B models as GenPRMs (replacing the 72B judge also to demonstrate efficiency). We also include more baselines like RLVR approaches RLOO [7] and using well trained process reward model (PRM) for credit assignment [8].
> * **Overall Results**: As shown in **Tables 1, 2, 3, and 4**, CAPO consistently outperforms GRPO, RLOO, and the PRM-based method even when using these smaller, more efficient GenPRMs. This demonstrates that CAPO does not require prohibitive compute to be effective; the fine-grained signal provided by a moderately sized judge (e.g., 14B) is sufficient to surpass rule-based baselines. **This indicated that our method can generalize to smaller judges and across various LLMs.**
>
> As for the new experiments applying CAPO directly to the **Qwen2.5-14B model** shown in the **Table 1**, CAPO consistently outperforms the GRPO baseline on the 14B scale. CAPO improved upon GRPO in math benchmark with avg 2.79% and 2.37% in OOD benchmark. By demonstrating consistent gains across 1.5B, 7B, and now 14B parameter scales, we have empirically verified that CAPO is a generalized credit assignment framework capable of enhancing models of varying sizes, dispelling the concern that its effectiveness is restricted to smaller architectures.

---

> ### Author Response · Authors · 2025-11-20
> **Rebuttal by Authors (continued)**
>
> #### **Table 1: Performance comparison across different methods using Qwen2.5-14B**. *CAPO uses Qwen2.5-32B-Instruct as a judge.*
> | model | gpqa | arc_c | mmlu_pro | OOD_Mean | math | olympiad_bench | minerva | aime24 | amc | aime25 | Math_Mean | All_Mean |
> |:---|:---:|:---:|:---:|:---:|:---:|:---:|:---:|:---:|:---:|:---:|:---:|:---:|
> | Base | 25.76 | 76.54 | 54.89 | 52.39 | 61.60 | 29.19 | 25.37 | 10.00 | 33.85 | 5.21 | 27.53 | 35.82 |
> | GRPO | 30.30 | 86.18 | 61.96 | 59.48 | **81.40** | **44.59** | 38.60 | 16.67 | 50.45 | 5.63 | 39.56 | 46.20 |
> | CAPO | **30.81** | **90.19** | **62.06** | **61.02** | 80.00 | 42.96 | **41.91** | **22.81** | **52.97** | **13.44** | **42.35** | **48.57** |
>
> ---
>
> #### **Table 2: Performance Comparison across different methods using Llama3.2-1B-Instruct.** *CAPO uses Qwen2.5-14B-Instruct as a judge.*
> | model | gpqa | arc_c | mmlu_pro | OOD_Mean | math | olympiad_bench | amc | gsm8k | Math_Mean | All_Mean |
> |:---|:---:|:---:|:---:|:---:|:---:|:---:|:---:|:---:|:---:|:---:|
> | Base | 18.69 | 6.91 | 1.30 | 8.96 | 22.60 | 4.89 | 5.65 | 20.62 | 13.44 | 11.52 |
> | GRPO | 20.54 | 51.68 | 17.16 | 29.79 | 31.00 | **8.44** | 11.23 | 51.28 | 25.49 | 27.33 |
> | RLOO | **24.41** | 32.99 | 16.39 | 24.60 | 30.13 | 5.63 | 14.48 | 48.42 | 24.67 | 24.64 |
> | PRM | 14.98 | 52.59 | 16.58 | 28.05 | 31.87 | 7.01 | 12.58 | 51.07 | 25.63 | 26.67 |
> | CAPO | 17.60 | **54.78** | **17.56** | **29.98** | **33.13** | 8.30 | **17.41** | **51.55** | **27.60** | **28.62** |
>
> ---
>
> #### **Table 3: Performance comparison across different methods using Qwen2.5-Math-1.5B.** *CAPO uses Qwen2.5-14B-Instruct as a judge.*
> | model | gpqa | arc_c | mmlu_pro | OOD_Mean | math | olympiad_bench | minerva | aime | amc | Math_Mean | All_Mean |
> |:---|:---:|:---:|:---:|:---:|:---:|:---:|:---:|:---:|:---:|:---:|:---:|
> | Base | 5.05 | 18.20 | 10.30 | 11.18 | 63.13 | 29.43 | 12.25 | 9.97 | 41.57 | 31.27 | 23.74 |
> | GRPO | **14.65** | **50.09** | 23.48 | **29.40** | 76.40 | 38.32 | 31.62 | 6.67 | 42.31 | 39.06 | 35.44 |
> | RLOO | 13.47 | 39.11 | 21.60 | 24.73 | 75.60 | 35.70 | 29.66 | 6.67 | 49.75 | 39.48 | 33.94 |
> | PRM | 10.61 | 45.88 | 22.99 | 26.49 | 76.00 | 38.81 | **32.35** | 12.40 | 43.30 | 40.57 | 35.29 |
> | CAPO | 13.13 | 43.34 | **24.28** | 26.92 | **76.80** | **40.20** | 32.23 | 18.19 | 44.73 | 42.43 | **36.61** |
> | CAPO_GenRM32B 1/3 steps | 13.64 | 41.81 | 20.46 | 25.30 | 74.70 | 38.00 | 28.86 | **23.18** | **49.85** | **42.92** | 36.31 |
>
> ---
>
> #### **Table 4: Performance comparison across different methods using Qwen2.5-Math-7B.** *CAPO uses Qwen2.5-14B-Instruct as a judge.*
> | model | gpqa | arc_c | mmlu_pro | OOD_Mean | math | olympiad_bench | minerva | aime | amc | Math_Mean | All_Mean |
> |:---|:---:|:---:|:---:|:---:|:---:|:---:|:---:|:---:|:---:|:---:|:---:|
> | Base | 18.18 | 30.32 | 16.71 | 21.74 | 64.73 | 19.80 | 9.07 | 11.35 | 42.28 | 29.45 | 26.56 |
> | GRPO | **25.76** | 64.11 | **44.62** | **44.83** | 81.13 | 44.10 | 36.76 | 30.00 | 56.78 | 49.75 | 47.91 |
> | RLOO | 22.22 | 42.43 | 41.84 | 35.50 | 79.33 | 41.73 | 33.95 | **40.52** | 59.46 | 51.00 | 45.19 |
> | PRM | 20.03 | **70.73** | 42.08 | 44.28 | 81.60 | 43.11 | **38.36** | 24.34 | **61.02** | 49.69 | 47.66 |
> | CAPO | 23.89 | 65.85 | 43.20 | 44.31 | **81.67** | **45.14** | 38.11 | 35.00 | 59.10 | **51.80** | **48.99** |
>
> ---

---

> ### Author Response · Authors · 2025-11-20
> **Rebuttal by Authors (continued)**
>
> ### **W3: Validation of the Motivation to Improve Reasoning Process Quality**
> **R3:** Thank you for this critical comment. We agree that demonstrating an actual improvement in the quality of the reasoning process—beyond just final accuracy—is essential to support our core hypothesis. We have added a comprehensive process-level error analysis.
>
> Conducting a process judge on full test set (spanning 19,425 samples across Math and OOD benchmarks, with contexts exceeding 10k tokens) is prohibitively expensive using outer APIs. We employed Qwen2.5-72B-Instruct as a highly capable judge to inspect the reasoning paths generated by the models trained in experiments (**specifically, the GRPO and CAPO models from Tables 5-7 are trained using the Simple-RL[6] setting, and with a 14B judge for CAPO instead of 72B in the original paper**).
>
> To rigorously quantify the quality of the reasoning process, we define two metrics: **Step Error Ratio (SER)**: The percentage of reasoning steps identified as incorrect and **Token Error Ratio(TER)**: The percentage of total tokens belonging to incorrect steps.
> Let dataset $\mathcal{D}$ contain $N$ generated responses. For the $i$-th response $Y_i$, let it consist of $K_i$ steps $(s_{i,1}, \dots, s_{i,K_i})$, where $|s_{i,k}|$ denotes the token count of step $s_{i,k}$, and $\mathbb{I}(s_{i,k})$ is the error indicator ($1$ if incorrect, $0$ otherwise). We calculate the metrics by averaging the error rates per response:
>
> \begin{equation}
> \text{SER} = \frac{1}{N} \sum_{i=1}^{N} \left( \frac{\sum_{k=1}^{K_i} \mathbb{I}(s_{i,k})}{K_i} \right)
> \end{equation}
>
> \begin{equation}
> \text{TER} = \frac{1}{N} \sum_{i=1}^{N} \left( \frac{\sum_{k=1}^{K_i} \mathbb{I}(s_{i,k}) \cdot |s_{i,k}|}{\sum_{k=1}^{K_i} |s_{i,k}|} \right)
> \end{equation}
>
>  This macro-averaging approach ensures that short and long reasoning chains are weighted equally, providing an unbiased evaluation of model performance across varying problem complexities.
>
>
> As shown in the **Table5-7** below, CAPO demonstrates a **significant reduction in process errors compared to GRPO** across both In-Domain (Math) and Out-Of-Domain benchmarks and achieve an average reduce up to 7%. The lower Error Token Ratio confirms that CAPO generates "cleaner" reasoning chains with fewer logic or calculation errors. These findings directly validate the motivation of CAPO. By providing reliable, fine-grained credit assignment, CAPO **effectively penalizes specific erroneous steps during training**. Unlike GRPO, which may reward a correct answer derived from flawed reasoning, CAPO forces the model to attend to the correctness of each intermediate step, thereby successfully shifting the learning objective from purely outcome-oriented to process-oriented.
>
> ---
>
> #### **Table 5: Comparison of step error ratio and token error ratio across different methods using Llama3.2-1B-Instruct**
> | Model | aime (SER) | aime (TER) | amc (SER) | amc (TER) | arc_c (SER) | arc_c (TER) | gpqa (SER) | gpqa (TER) | math (SER) | math (TER) | minerva (SER) | minerva (TER) | mmlu_pro (SER) | mmlu_pro (TER) | olympiad (SER) | olympiad (TER) |  OOD_Mean (SER) |  OOD_Mean (TER) | Math_Mean (SER) | Math_Mean (TER) |
> | :--- | :---: | :---: | :---: | :---: | :---: | :---: | :---: | :---: | :---: | :---: | :---: | :---: | :---: | :---: | :---: | :---: | :---: | :---: | :---: | :---: |
> | GRPO | 67.29 | 67.66 | 52.06 | 52.83 | 25.22 | 22.80 | 55.65 | 59.41 | 40.31 | 38.40 | 56.56 | 57.73 | 47.33 | 45.30 | 56.49 | 56.63 | 42.73 | 42.50 | 54.54 | 54.65 |
> | CAPO | **51.10** | **50.41** | **49.01** | **48.76** | **23.06** | **21.19** | **49.02** | **49.43** | **37.95** | **34.71** | **55.54** | **54.46** | **45.73** | **41.59** | **51.72** | **49.94** | **39.27** | **37.40** | **49.06** | **47.66** |
>
> ---
>
> #### **Table 6: Comparison of step error ratio and token error ratio across different methods using Qwen2.5-Math-1.5B**
> | Model | aime (SER) | aime (TER) | amc (SER) | amc (TER) | arc_c (SER) | arc_c (TER) | gpqa (SER) | gpqa (TER) | math (SER) | math (TER) | minerva (SER) | minerva (TER) | mmlu_pro (SER) | mmlu_pro (TER) | olympiad (SER) | olympiad (TER) |  OOD_Mean (SER) |  OOD_Mean (TER) | Math_Mean (SER) | Math_Mean (TER) |
> | :--- | :---: | :---: | :---: | :---: | :---: | :---: | :---: | :---: | :---: | :---: | :---: | :---: | :---: | :---: | :---: | :---: | :---: | :---: | :---: | :---: |
> | GRPO | 27.62 | 41.11 | 19.64 | 23.99 | 23.66 | 23.74 | 37.74 | 48.01 | 10.93 | 11.12 | 19.96 | 24.94 | 33.43 | 33.03 | 22.62 | 26.34 | 31.61 | 34.93 | 20.15 | 25.50 |
> | CAPO | **26.01** | **34.93** | **13.01** | **14.87** | **20.60** | **20.16** | **36.31** | **44.55** | **7.45** | **8.06** | **15.52** | **19.18** | **27.41** | **28.32** | **17.79** | **18.39** | **28.11** | **31.01** | **15.96** | **19.09** |

---

> > ### Comment · Reviewer_RfL9 · 2025-11-27
> >
> > How was the "error indicator for each step" implemented?

---

> > > ### Author Response · Authors · 2025-11-27
> > > **Reponse to Further Question**
> > >
> > > Thanks for your further question. We further elaborate the detailed implementation as below:
> > >
> > > As outlined in our response to **W3**, we employed **Qwen2.5-72B-Instruct** as a highly capable "Meta-Judge" to automate this process. For each evaluation sample, we first constructed a prompt containing the **[Question]**, the **[Ground Truth Solution]**, and the **[Solution To Judge]** using the prompt template below. Crucially, we separated the steps using \n\n and pre-processed the solution by wrapping each step with index tags (e.g., `<step_i>...</step_i>`) to allow precise referencing.
> > > We used the prompt to instruct the judge to verify the reasoning chain, and the judge then outputs the indices of erroneous steps in a structured format.
> > >
> > > We parsed the judge's output (specifically the `<incorrect_steps>` block). **If index $k$ appears in the list, the error indicator $\mathbb{I}(s_k)$ is set to **1**; otherwise, it is **0**. These indicators are then used to calculate the SER and TER.**
> > >
> > > The exact prompt used in our implementation is provided below:
> > >
> > > ```text
> > > You are a math teacher. Use [Ground Truth Solution] to find any erroneous step in [Solution To Judge]. And the solution to be checked is separated with steps like "<step_i>...</step_i>" where i is the index of the step.
> > >
> > > First, determine whether the solution is correct based on [Ground Truth Solution] and write it in the form "Judgement: Is the solution correct (Yes/No)? X", where X is either Yes or No.
> > >
> > > If the solution is incorrect, review, analyze, and verify each step of the solution independently, without relying on the correctness of the context before. This means that the current step cannot be considered as incorrect simply because it used the conclusion derived from the previous erroneous reasoning.
> > >
> > > Then provide the list of the indices of error steps separated by a comma. And the format is as follows:
> > > <incorrect_steps>
> > > ...(the index of the incorrect steps)...
> > > </incorrect_steps>.
> > >
> > > Please give your answer directly.
> > >
> > > [Question]
> > > {problem}
> > >
> > > [Ground Truth Solution]
> > > {gt_solution}
> > >
> > > [Solution To Judge]
> > > {solution}
> > >
> > > ```
> > >
> > > ---
> > >
> > > Thank you again for your detailed review. Please feel free to let us know if you have any further questions.

---

> ### Author Response · Authors · 2025-11-21
> **Rebuttal by Authors (continued)**
>
> #### **Table 7: Comparison of step error ratio and token error ratio across different methods using Qwen2.5-Math-7B**
> | Model | aime (SER) | aime (TER) | amc (SER) | amc (TER) | arc_c (SER) | arc_c (TER) | gpqa (SER) | gpqa (TER) | math (SER) | math (TER) | minerva (SER) | minerva (TER) | mmlu_pro (SER) | mmlu_pro (TER) | olympiad (SER) | olympiad (TER) |  OOD_Mean (SER) |  OOD_Mean (TER) | Math_Mean (SER) | Math_Mean (TER) |
> | :--- | :---: | :---: | :---: | :---: | :---: | :---: | :---: | :---: | :---: | :---: | :---: | :---: | :---: | :---: | :---: | :---: | :---: | :---: | :---: | :---: |
> | GRPO | 32.83 | 34.37 | 12.31 | 14.82 | 12.94 | 12.23 | 29.55 | 34.01 | **4.96** | **5.49** | 15.10 | 15.84 | 30.86 | 30.09 | 15.24 | 16.47 | 24.45 | 25.44 | 16.09 | 17.40 |
> | CAPO | **20.16** | **23.17** | **10.46** | **12.38** | **7.44** | **6.42** | **24.76** | **30.50** | 5.98 | 5.82 | **10.72** | **13.15** | **20.29** | **18.94** | **13.93** | **16.03** | **17.49** | **18.62** | **12.25** | **14.11** |
>
> ---
>
> ### **W4: Typos Issues**
> **R4:** We sincerely thank you for your careful reading and attention to detail! We have corrected the capitalization error on line 189 as suggested ("r" to "R"). Additionally, we have thoroughly proofread the entire manuscript to ensure that all similar editorial issues are resolved in the final version.
>
> ---
>
> ### **Q1: The group normalization of process reward**
> **R1:** Thank you for the thoughtful question. We believe that group-relative normalization is both theoretically sound and empirically effective for process supervision. Our implementation follows the widely adopted setting in the RLVR community. Specifically, we follow [5], which explicitly discusses applying GRPO to process supervision in Section 4.1.3. As demonstrated in their work and ours, applying group normalization to process rewards is a standard technique to estimate the baseline (advantage function) without training a separate critic model. Theoretically, this normalization acts as a robust variance reduction technique, similar to the standard GRPO for outcome rewards. It is important to note that reasoning steps across different rollouts are often **misaligned**. Therefore, a global group baseline is more reliable than a step-specific one. Furthermore, since normalization is a linear transformation, it preserves the **relative ranking** of the rewards. High-quality steps still receive high positive advantages relative to the group mean, ensuring that the fine-grained, step-specific signal is effectively captured and not obscured.
>
> ---
>
> [1] Zhang, Zhenru, et al. "The lessons of developing process reward models in mathematical reasoning." ACL2025
>
> [2] Zheng, Chujie, et al. "Processbench: Identifying process errors in mathematical reasoning." ACL2025
>
> [3] Yang, An, et al. "Qwen2. 5-math technical report: Toward mathematical expert model via self-improvement." arxiv 2024.
>
> [4] Liu, Yuliang, et al. "Adaptivestep: Automatically dividing reasoning step through model confidence." arXiv 2025
>
> [5] Shao, Zhihong, et al. "Deepseekmath: Pushing the limits of mathematical reasoning in open language models." arxiv 2024.
>
> [6] Zeng, Weihao, et al. "Simplerl-zoo: Investigating and taming zero reinforcement learning for open base models in the wild." COLM 2025
>
> [7] Ahmadian, Arash, et al. "Back to basics: Revisiting reinforce style optimization for learning from human feedback in llms." ACL2024
>
> [8] Cheng, Jie, et al. "Stop summation: Min-form credit assignment is all process reward model needs for reasoning." NeurIPS 2025

---

### Official Review · Reviewer_p6qQ · 2025-11-01

**Soundness:** 3
**Presentation:** 3
**Contribution:** 3
**Rating:** 4
**Confidence:** 3

**Summary:**

This paper identifies a key limitation in Reinforcement Learning with Verifiable Rewards (RLVR) for fine-tuning Large Language Models (LLMs): the coarse-grained, binary (correct/incorrect) reward assigned to an entire response lacks the granularity to provide differentiated feedback on individual reasoning steps. This imprecise credit assignment hinders the model's ability to discern which specific parts of its reasoning are sound or flawed. To address this, the authors propose Credit Assignment Policy Optimization (CAPO), a method that introduces fine-grained, token-level credit assignment, avoiding the complexity and potential unreliability of training auxiliary Process Reward Models (PRMs).
Extensive experiments on mathematical and general reasoning benchmarks, using the Llama and Qwen model families (1B-7B), show that CAPO consistently outperforms both supervised fine-tuning and RL baselines like GRPO. Analyses indicate that CAPO leads to longer, more exploratory responses that correlate with higher final accuracy, suggesting it effectively fosters correct reasoning pathways.

**Strengths:**

The paper presents a creative synthesis of existing ideas. Using a general-purpose LLM as a verifier for process supervision is a simple yet powerful insight that effectively bypasses the significant overhead of training dedicated PRMs while leveraging the inherent reasoning capabilities of modern LLMs. The formulation of the credit assignment mechanism and the analysis of the outcome-process reward trade-off are novel contributions to the RLVR framework.

**Weaknesses:**

While efficient, the method introduces non-trivial computational overhead compared to simpler rule-based verifiers such as GRPO. Each policy rollout requires multiple inference passes with a very large LLM (GenPRM). Although the paper correctly notes this aligns with trends of using large models as guides, a more detailed discussion of the actual cost—such as GPU hours or a direct wall-clock time comparison with GRPO—would strengthen the practicality claim. The paper is transparent about this cost, but a deeper analysis would be valuable.

The approach assumes that a significantly more capable GenPRM provides reliable feedback. While reasonable, this assumption introduces an inherent scalability ceiling; the method's effectiveness might diminish as the policy model's capability approaches that of the judge, a scenario not explored here. This relates to the broader, unexamined question of the quality of the process supervision provided by the LLM-judge itself.

**Questions:**

Please refer to "Weaknesses".

---

> ### Author Response · Authors · 2025-11-20
> **Rebuttal by Authors**
>
> Thank you for your careful and insightful feedback! We appreciate that you found our credit assignment mechanism novel and creative, and the analysis of the outcome-process reward trade-off as contributions to the RLVR framework. Below, we address the weaknesses you have raised:
>
> ***Note on Code Update: We have updated the supplementary materials and the anonymous link with a new implementation based on the widely adopted verl framework.*** This replaces the previous version, which relied on a proprietary internal RL infrastructure, to ensure better **accessibility and reproducibility for the community**.
>
> ### **W1: The computational cost of CAPO**
> **R1:** Thank you for your comment and we fully agree that a transparent discussion on the cost of using GenPRMs is necessary. We have addressed this by conducting extensive new experiments with smaller GenPRMs and providing a detailed breakdown of GPU hours.
>
> **1. Robustness with Smaller GenPRMs**
>
> To mitigate the concern regarding the reliance on "very large" models (e.g., the 72B model used in our initial draft), we have significantly expanded our experiments using more efficient models as judges, using 14B model as backbones and including more baselines.
> * **Experimental Setup**: Following the standard Simple-RL [1] setting, we conducted extensive experiments across various sizes including 1B, 1.5B, 7B, and 14B models. We employed **Qwen2.5-14B-Instruct** for 1B, 1.5B and 7B models and **Qwen2.5-32B-Instruct** for 14B models as GenPRMs, rather than just 72B models. We also include more baselines like RLVR approaches RLOO [3] and using well trained process reward model (PRM) for credit assignment [2].
> * **Results**: As shown in **Tables 1, 2, 3, and 4**, CAPO consistently outperforms GRPO, RLOO, and the PRM-based method even when using these smaller, more efficient GenPRMs. This demonstrates that CAPO does not require prohibitive compute to be effective; the fine-grained signal provided by a moderately sized judge (e.g., 14B) is sufficient to surpass rule-based baselines.
>
> **2. Quantified Computational Overhead: CAPO vs. GRPO**
>
> We also provide and analyze the GPU hours of training Qwen2.5-Math-7B using a locally deployed Qwen2.5-14B-Instruct judge via vllm (**the cost of deployment is also included in the compuatation of GPU hours below**) and that using GRPO.
>
> * Rule-based Verification (GRPO): While simpler, this is not cost-free due to the overhead of external parsing libraries (e.g., math-verify) which lack thread safety. The average reward evaluation is **4.5s**, with a total per-step time of **~47s**.
> * CAPO: The average reward evaluation is **12.1s**, with a total per-step time of **~55s**.
> * Total Cost: For a full training cycle, GRPO consumed **272 GPU hours**, while CAPO consumed **331.2 GPU hours**. This represents a **~21.7% increase**, which we believe is a manageable overhead given the performance gains and the reasons are as follows.
>
> **3. Efficiency of CAPO via Faster Convergence and Higher Ceilings**
>
> As detailed in **Table 5**, we showed the performance of GRPO and CAPO in different steps. We found that GRPO converge relatively slow and with limited gains on extended training. CAPO, however, achieve a high performance in a very beginning (step 60 and step 90) compared to GRPO which is also higher than the final training of GRPO and continue to improve. It is important to note that the 21.7% increase of GPU hours is based on the entire training but we actually can reduce the training steps for CAPO like using half steps (90 steps). In this situation, we still perform better than GRPO and using comparable or less GPU hours to GRPO.
>
> We also show the further potential of CAPO to accelerate convergence. In our Qwen2.5-Math-1.5B experiments (**Table 2**), using a stronger judge (32B) allowed the model to reach optimal performance in just **1/3 of the training steps** required by the baseline. By enabling early stopping or reduce the total training steps, CAPO can, in practice, **reduce the total GPU hours** compared to rule-based methods to convergence, effectively offsetting the per-step overhead.
>
> ---
>
> #### **Table 1: Performance Comparison across different methods using Llama3.2-1B-Instruct.** *CAPO uses Qwen2.5-14B-Instruct as a judge.*
> | model | gpqa | arc_c | mmlu_pro | OOD_Mean | math | olympiad_bench | amc | gsm8k | Math_Mean | All_Mean |
> |:---|:---:|:---:|:---:|:---:|:---:|:---:|:---:|:---:|:---:|:---:|
> | Base | 18.69 | 6.91 | 1.30 | 8.96 | 22.60 | 4.89 | 5.65 | 20.62 | 13.44 | 11.52 |
> | GRPO | 20.54 | 51.68 | 17.16 | 29.79 | 31.00 | **8.44** | 11.23 | 51.28 | 25.49 | 27.33 |
> | RLOO | **24.41** | 32.99 | 16.39 | 24.60 | 30.13 | 5.63 | 14.48 | 48.42 | 24.67 | 24.64 |
> | PRM | 14.98 | 52.59 | 16.58 | 28.05 | 31.87 | 7.01 | 12.58 | 51.07 | 25.63 | 26.67 |
> | CAPO | 17.60 | **54.78** | **17.56** | **29.98** | **33.13** | 8.30 | **17.41** | **51.55** | **27.60** | **28.62** |

---

> ### Author Response · Authors · 2025-11-20
> **Rebuttal by Authors (continued)**
>
> #### **Table 2: Performance comparison across different methods using Qwen2.5-Math-1.5B.** *CAPO uses Qwen2.5-14B-Instruct as a judge.*
> | model | gpqa | arc_c | mmlu_pro | OOD_Mean | math | olympiad_bench | minerva | aime | amc | Math_Mean | All_Mean |
> |:---|:---:|:---:|:---:|:---:|:---:|:---:|:---:|:---:|:---:|:---:|:---:|
> | Base | 5.05 | 18.20 | 10.30 | 11.18 | 63.13 | 29.43 | 12.25 | 9.97 | 41.57 | 31.27 | 23.74 |
> | GRPO | **14.65** | **50.09** | 23.48 | **29.40** | 76.40 | 38.32 | 31.62 | 6.67 | 42.31 | 39.06 | 35.44 |
> | RLOO | 13.47 | 39.11 | 21.60 | 24.73 | 75.60 | 35.70 | 29.66 | 6.67 | 49.75 | 39.48 | 33.94 |
> | PRM | 10.61 | 45.88 | 22.99 | 26.49 | 76.00 | 38.81 | **32.35** | 12.40 | 43.30 | 40.57 | 35.29 |
> | CAPO | 13.13 | 43.34 | **24.28** | 26.92 | **76.80** | **40.20** | 32.23 | 18.19 | 44.73 | 42.43 | **36.61** |
> | CAPO_GenRM32B 1/3 steps | 13.64 | 41.81 | 20.46 | 25.30 | 74.70 | 38.00 | 28.86 | **23.18** | **49.85** | **42.92** | 36.31 |
>
> ---
>
> #### **Table 3: Performance comparison across different methods using Qwen2.5-Math-7B.** *CAPO uses Qwen2.5-14B-Instruct as a judge.*
> | model | gpqa | arc_c | mmlu_pro | OOD_Mean | math | olympiad_bench | minerva | aime | amc | Math_Mean | All_Mean |
> |:---|:---:|:---:|:---:|:---:|:---:|:---:|:---:|:---:|:---:|:---:|:---:|
> | Base | 18.18 | 30.32 | 16.71 | 21.74 | 64.73 | 19.80 | 9.07 | 11.35 | 42.28 | 29.45 | 26.56 |
> | GRPO | **25.76** | 64.11 | **44.62** | **44.83** | 81.13 | 44.10 | 36.76 | 30.00 | 56.78 | 49.75 | 47.91 |
> | RLOO | 22.22 | 42.43 | 41.84 | 35.50 | 79.33 | 41.73 | 33.95 | **40.52** | 59.46 | 51.00 | 45.19 |
> | PRM | 20.03 | **70.73** | 42.08 | 44.28 | 81.60 | 43.11 | **38.36** | 24.34 | **61.02** | 49.69 | 47.66 |
> | CAPO | 23.89 | 65.85 | 43.20 | 44.31 | **81.67** | **45.14** | 38.11 | 35.00 | 59.10 | **51.80** | **48.99** |
>
> ---
>
> #### **Table 4: Performance comparison across different methods using Qwen2.5-14B.** *CAPO uses Qwen2.5-32B-Instruct as a judge.*
> | model | gpqa | arc_c | mmlu_pro | OOD_Mean | math | olympiad_bench | minerva | aime24 | amc | aime25 | Math_Mean | All_Mean |
> |:---|:---:|:---:|:---:|:---:|:---:|:---:|:---:|:---:|:---:|:---:|:---:|:---:|
> | Base | 25.76 | 76.54 | 54.89 | 52.39 | 61.60 | 29.19 | 25.37 | 10.00 | 33.85 | 5.21 | 27.53 | 35.82 |
> | GRPO | 30.30 | 86.18 | 61.96 | 59.48 | **81.40** | **44.59** | 38.60 | 16.67 | 50.45 | 5.63 | 39.56 | 46.20 |
> | CAPO | **30.81** | **90.19** | **62.06** | **61.02** | 80.00 | 42.96 | **41.91** | **22.81** | **52.97** | **13.44** | **42.35** | **48.57** |
>
> ---
>
> #### **Table 5: Performance comparison of different steps using Qwen2.5-Math-7B.** *CAPO uses Qwen2.5-14B-Instruct as a judge.*
> | model_name | step | gpqa | arc_c | mmlu_pro | OOD_Mean | math | olympiad_bench | minerva | aime | amc | Math_Mean | All_Mean |
> |:---|:---:|:---:|:---:|:---:|:---:|:---:|:---:|:---:|:---:|:---:|:---:|:---:|
> | Base | | 18.18 | 30.32 | 16.71 | 21.74 | 64.73 | 19.80 | 9.07 | 11.35 | 42.28 | 29.45 | 26.56 |
> | GRPO | 60 | 23.23 | 59.44 | 43.69 | 42.12 | 80.53 | 41.53 | 38.73 | 26.67 | 61.38 | 49.77 | 46.90 |
> | | 90 | 22.90 | 65.22 | 44.70 | 44.27 | 79.33 | 40.89 | 38.85 | 28.96 | 50.62 | 47.73 | 46.43 |
> | | 120 | 24.75 | 60.75 | 44.69 | 43.40 | 81.93 | 43.56 | 34.56 | 24.17 | 60.17 | 48.88 | 46.82 |
> | | 176 | 25.76 | 64.11 | 44.62 | 44.83 | 81.13 | 44.10 | 36.76 | 30.00 | 56.78 | 49.75 | 47.91 |
> | CAPO | 60 | 22.89 | 58.06 | 43.94 | 41.63 | 81.33 | 43.17 | 38.03 | 28.11 | 60.49 | 50.23 | 47.00 |
> | | 90 | 20.01 | 60.89 | 43.94 | 41.61 | 82.27 | 43.11 | 38.97 | 30.00 | 60.99 | 51.07 | 47.52 |
> | | 120 | 23.89 | 65.85 | 43.20 | 44.31 | 81.67 | 45.14 | 38.11 | 35.00 | 59.10 | 51.80 | 48.99 |
> | | 176 | 23.74 | 69.23 | 39.57 | 44.18 | 80.33 | 40.30 | 37.38 | 26.56 | 50.94 | 47.10 | 46.01 |

---

> ### Author Response · Authors · 2025-11-20
> **Rebuttal by Authors (continued)**
>
> ### **W2: On the Dependency on a More Capable GenPRM**
> **R2:** Thank you for raising this important point regarding the scalability of our approach and the "weak-to-strong" generalization gap. While we agree that all guidance-based paradigms (whether PRMs, Critics, or Teachers) benefit from stronger models, we offer three key arguments supported by new experiments to address the concern about the "scalability ceiling."
>
> **1. Reducing Dependency via Ground Truth (Verification is Easier than Generation)**
>
> A critical design choice in CAPO effectively lowers the capability requirement for the GenPRM: **access to the Ground Truth**. Unlike a standard solver that must reason from scratch, the GenPRM in CAPO is conditioned on the correct final answer during the critique phase. This shifts the task from generation to verification which is easier than generating it from scratch. Therefore, we don't need a significantly strong model to be a judge.
>
> **2. Empirical Evidence with Smaller Judges**
>
> As also seen in response to W1, we conducted extensive new experiments using **Qwen2.5-14B-Instruct** as the GenPRM. As shown in **Tables 1, 2, 3, and 4**, CAPO driven by a 14B judge consistently outperforms baselines and achieves gains comparable to those obtained with a 72B judge.
>
> **3. Accessibility**
>
> Finally, we highlight a practical advantage. Traditional PRM-based methods often require expensive additional training (costly data collection + model fine-tuning) to create a capable verifier. In contrast, CAPO leverages off-the-shelf, general-purpose models (even smaller ones like 14B) without any additional training. This makes the method highly accessible and scalable, as it bypasses the need to train a "super-judge" from scratch.
>
> ---
>
> [1] Zeng, Weihao, et al. "Simplerl-zoo: Investigating and taming zero reinforcement learning for open base models in the wild." COLM 2025
>
> [2] Cheng, Jie, et al. "Stop summation: Min-form credit assignment is all process reward model needs for reasoning." NeurIPS 2025
>
> [3] Ahmadian, Arash, et al. "Back to basics: Revisiting reinforce style optimization for learning from human feedback in llms." ACL2024

---

### Comment · Area_Chair_M8sh · 2025-11-21
**Action Required: Review of Author Rebuttal**

Dear Reviewers,

The authors have submitted their rebuttal. Please review their responses and provide any follow-up, such as additional questions or revisions to your review.

The reviewers have expressed differing views, and we note that Reviewer zJU4 has raised their score based on the rebuttal. As you form your final recommendation, we ask that you give careful consideration to both the authors' response and the perspectives of your fellow reviewers.

Thank you for your contributions to this process.

Sincerely,
Your AC

---

### Meta-Review · Area_Chair_dXcj · 2026-01-02

**Summary:**

This paper proposes credit assignment policy optimization, CAPO, which directly adopts an off-the-shelf general purpose LLM as a generative process reward model, to produce deterministic token-level credits and overcome the drawbacks of rule-based identical rewards or training auxiliary reward models. Experiments across various models and benchmarks are conducted to verify the performance.


The major concerns of reviewers include: (1) additional challenge and computational cost and lack of analysis regarding the overhead compared to rule-based verifiers, as raised by Reviewers p6qQ and NbTj; (2) scalability due to the reliance on capable GenPRM, as raised by Reviewers p6qQ and zJU4; (3) limited performance improvement and only verification on models with less than 10B parameters, as raised by Reviewers RfL9 and NbTj; (4) justification for the segmentation strategy, as raised by Reviewers RfL9 and zJU4; (5) lack of analysis and verification for the motivation of improving reasoning quality, as raised by Reviewers RfL9 and NbTj.

**Reviewer Concerns:**

In the rebuttal process, one reviewer acknowledged that their concerns were addressed and changed the score from 6 to 8. However, after going through the paper and all the discussion comments, the AC considers some of the above concerns not fully resolved. For example, regarding the computational overhead, from the numbers provided by authors, the total training cost is increased from 272 GPU hours for GRPO to 331.2 GPU hours for CAPO, a >20% increase. In general, more results and analyses around these issues need to be complemented to enhance this work.

**Reviewer Scores:**

Reviewer p6qQ: 4

Reviewer RfL9: 4

Reivewer zJU4: 8

Reviewer NbTj: 4

---

### Decision · Program_Chairs · 2026-01-26

Reject